# Reprogramming of pancreatic adenocarcinoma immunosurveillance by a microbial probiotic siderophore

Mehdi Chaib[1], Bilal B. Hafeez[2✉], Hassan Mandil[1], Deidre Daria[3], Ajeeth K. Pingili[4], Sonam Kumari[1], Mohammed Sikander[2], Vivek K. Kashyap[2], Guo-Yun Chen[5], Emmanuel Anning[2], Manish K. Tripathi[2], Sheema Khan[2], Stephen Behrman[6], Murali M. Yallapu[2], Meena Jaggi[2], Liza Makowski[7] & Subhash C. Chauhan[2✉]

There is increasing evidence suggesting the role of microbiome alterations in relation to pancreatic adenocarcinoma and tumor immune functionality. However, molecular mechanisms of the interplay between microbiome signatures and/or their metabolites in pancreatic tumor immunosurveillance are not well understood. We have identified that a probiotic strain (*Lactobacillus casei*) derived siderophore (ferrichrome) efficiently reprograms tumor-associated macrophages (TAMs) and increases CD8 + T cell infiltration into tumors that paralleled a marked reduction in tumor burden in a syngeneic mouse model of pancreatic cancer. Interestingly, this altered immune response improved anti-PD-L1 therapy that suggests promise of a novel combination (ferrichrome and immune checkpoint inhibitors) therapy for pancreatic cancer treatment. Mechanistically, ferrichrome induced TAMs polarization *via* activation of the TLR4 pathway that represses the expression of iron export protein ferroportin (FPN1) in macrophages. This study describes a novel probiotic based molecular mechanism that can effectively induce anti-tumor immunosurveillance and improve immune checkpoint inhibitors therapy response in pancreatic cancer.

[1] Department of Pharmaceutical Sciences, College of Pharmacy, University of Tennessee Health Science Center (UTHSC), Memphis, TN 38163, USA. [2] Department of Immunology and Microbiology and South Texas Center of Excellence in Cancer Research, School of Medicine, University of Texas Rio Grande Valley, McAllen, TX 78504, USA. [3] Department of Microbiology, Immunology and Biochemistry, Memphis, TN 38163, USA. [4] Division of Hematology Oncology, Department of Medicine, Memphis, TN 38163, USA. [5] Children's Foundation Research Institute at Le Bonheur Children's Hospital, Department of Pediatrics, Memphis, TN 38163, USA. [6] Department of Surgery, Memphis, TN 38163, USA. [7] Department of Medicine, Division of Hematology and Oncology and the UTHSC Center for Cancer Research, Memphis, TN 38103, USA. ✉email: bilal.hafeez@utrgv.edu; subhash.chauhan@utrgv.edu

The immune surveillance of pancreatic tumors is highly compromised due to complex histoarchitecture, desmoplasia, and poor vascularization that ultimately cause immunoresistance[1]. Immunotherapy has received much attention in recent years and is standard of care for several cancers, yet their therapeutic benefits have not extended to pancreatic cancer[2]. Due to lack of effective diagnostic and therapeutic modalities, pancreatic cancer is going to be the second leading cause of cancer-related deaths by year 2030[3]. At present, the 5-year survival of this disease is less than nine percent and is the third leading cause of cancer-related deaths in the United States[4]. Although 10–15% of patients are candidates for surgical resection, standard therapies have only a marginal impact on survival[5,6], thus, development of novel therapeutic approaches is urgently warranted for this devastating disease.

Pancreatic tumors, like other solid tumors, are heavily infiltrated by inflammatory leukocytes, but their antitumor immune response is ineffective due to immunosuppression[7,8]. Strategies to reverse immune suppression by targeting myeloid cells *via* vaccination with pancreatic tumor antigens, or by application of immune checkpoint inhibitors suggest that proper targeting of the immune system may generate effective strategies to target pancreatic tumors[9]. Tumor associated macrophages (TAMs) represent a major component of immune infiltrating leukocytes in pancreatic tumors[10]. Soluble factors secreted by TAMs have been shown to promote cancer progression by establishing an immunosuppressive microenvironment that inhibits antitumor T-cell responses. TAMs also mediate resistance to conventional chemotherapies and contemporary targeted regimens[11–13]. Macrophages are highly plastic cells that acquire distinct phenotypes in response to signals present within different microenvironments[14–16]. In response to bacterial products such as lipopolysaccharide (LPS) and T helper 1 cytokines, macrophages are polarized to the classical or "M1"-like phenotype, leading to the expression of proinflammatory cytokines[17]. M1 macrophages are characterized by high expression of inducible nitric oxide synthase (iNOS), tumor necrosis factor α (TNF-α), CD80, CD86, major histocompatibility complex class II (MHCII) proteins, interleukins such as IL-6 and IL-12, and can exert an antitumor effect[18]. In contrast, activated "M2"-like macrophages can be polarized in the presence of Th2 cytokines such as IL-4/IL-13, IL-10, or transforming growth factor beta (TGF-β). M2 macrophages are immunosuppressive and induce tumor-promoting functions by enhancing tissue remodeling and facilitating angiogenesis[19,20]. In pancreatic tumors, TAMs exhibit both M1 and M2 phenotypes, but show preferential expression of M2 markers such as Arginase 1, CD206, and TGF-β leading to a higher M2:M1 ratios which correlates with a poorer prognosis in pancreatic cancer patients[21,22]. Macrophages that are the vanguard of innate immune system also play a pivotal role in iron homeostasis[23–25]. Macrophage polarization is highly dependent on iron metabolism, which becomes deregulated in tumors, thus affects the tumor immunity[26]. Studies have shown that over 60% of iron metabolism-related genes are differentially expressed between the M1/M2 polarization axis[27]. M1 macrophages express high levels of the iron storage protein ferritin and low levels of iron export protein ferroportin (Fpn). In contrast, M2 macrophages display an iron-export phenotype through increased expression of *Fpn* and decreased expression of ferritin leading to iron export and little storage[27–30]. Therefore, higher iron availability in the tumor microenvironment (TME) was observed which provided growing cancer cells with iron critical to enzymatic processes and tumor growth[31,32]. Fpn concentration is regulated by increased expression of hepcidin (HAMP) which causes Fpn internalization and degradation. Endogenous expression of HAMP by macrophages is increased upon activation of Toll-like Receptor 4 (TLR4) by bacterial products such as LPS. HAMP binds to Fpn and causes Fpn internalization and degradation leading to

an iron storage-associated pro-inflammatory M1 phenotype[33–36]. In recent years, a link between the human microbiome and pancreatic cancer has been established and altered patterns of oral and gut microbiome can increase the risk of pancreatic cancer[37,38]. Some probiotic microbiota species, however, can provide beneficial effects to prevent the tumor growth[39,40]. Recent investigations have shown tumor-suppressive potential of ferrichrome, a siderophore (iron chelator), produced by the probiotic strain *Lactobacillus casei* in colon and gastric cancers[40,41].

In this study, we report that ferrichrome regulates macrophage polarization in a TLR4-dependent manner by promoting macrophage transcription of pro-inflammatory mediators, thus preventing immunosuppressive TAMs polarization. Moreover, we elucidated that ferrichrome regulates macrophage iron metabolism by decreasing Fpn expression in a TLR4-dependent manner, not by iron chelation. Using in vitro mono and co-culture systems and in vivo syngeneic mouse models, we show that ferrichrome promotes the M1 phenotype and abrogates the M2 phenotype. Furthermore, ferrichrome generates an immune-responsive microenvironment with increased presence of CD8 + T cells and decreased myeloid suppressive populations. Surprisingly, ferrichrome drastically increased responsiveness of anti-PD-L1 antibody therapy in pancreatic tumors, suggesting that pancreatic cancer patients expressing PDL-1 may be more responsive to immune checkpoint blockade (ICB) therapies in the presence of ferrichrome. Our findings suggest ferrichrome as a novel TLR4 agonist that could be used as an adjuvant for the treatment of pancreatic cancer and other solid tumors. Taken together, our findings illustrate an innovative concept that a probiotic-derived siderophore can reprogram the pancreatic TME through TLR4, which is permissive to improve tumor immune surveillance and responsiveness to immune checkpoint inhibitors immunotherapies.

## Results

**Ferrichrome attenuates growth of pancreatic tumors.** Antitumorigenic effects of probiotic microbial species have been reported in recent studies[40,41]. To advance the knowledge regarding the influence of probiotics on tumor immunity, we elucidated if ferrichrome—(Fig. 1a, $C_{27}H_{42}FeN_9O_{12}$), a product of the probiotic strain *Lactobacillus casei*—could potentiate an antitumor immune response and attenuate pancreatic tumor growth. We used a pancreatic cancer syngeneic mouse model wherein the mouse pancreatic cancer cell line (UN-KC-6141) was injected into the flanks of female C57BL/6 J wild-type mice to establish KC tumor model as described in (Fig. 1b). Ferrichrome (50 μg/mouse intratumoral (IT), 5 days a week over ~3 weeks) inhibited tumor growth compared to vehicle-treated mice as evident by significant decrease in tumor volume ($p < 0.05$–$p < 0.001$) and tumor weight ($p < 0.01$) (Fig. 1c–e), while showing no apparent toxicity (Fig. 1f). Interestingly, ferrichrome had no growth inhibitory effect in two mouse pancreatic cancer cell lines up to 120 μM in vitro (Supplementary Fig. 1a–b), suggesting that the tumor growth arrest induced by ferrichrome may be caused by an antitumor immune response rather than a direct cytotoxic effect. Ferrichrome displayed moderate inhibition of proliferation in human colon cancer cell lines SW480 (Supplementary Fig. 1c–d) and SW620 (Supplementary Fig. 1e–f), consistent with previous reports[40]. Ferrichrome also had inhibitory growth effects on RAW264.7 murine macrophages (Supplementary Fig. 1g–h). However, ferrichrome has no significant effect on induction of apoptosis in BMDMs, UN-KPC-960, or SW620 cells (Supplementary Fig. 2) suggesting that ferrichrome does not cause cell death in cancer cells or in macrophages. Histological analysis of excised tumors using immunohistochemistry revealed a significant ($p < 0.01$) increase in CD8 + T cells, a decrease in CD163 cells (M2 TAMs marker), and a significant ($p < 0.001$) decrease in the

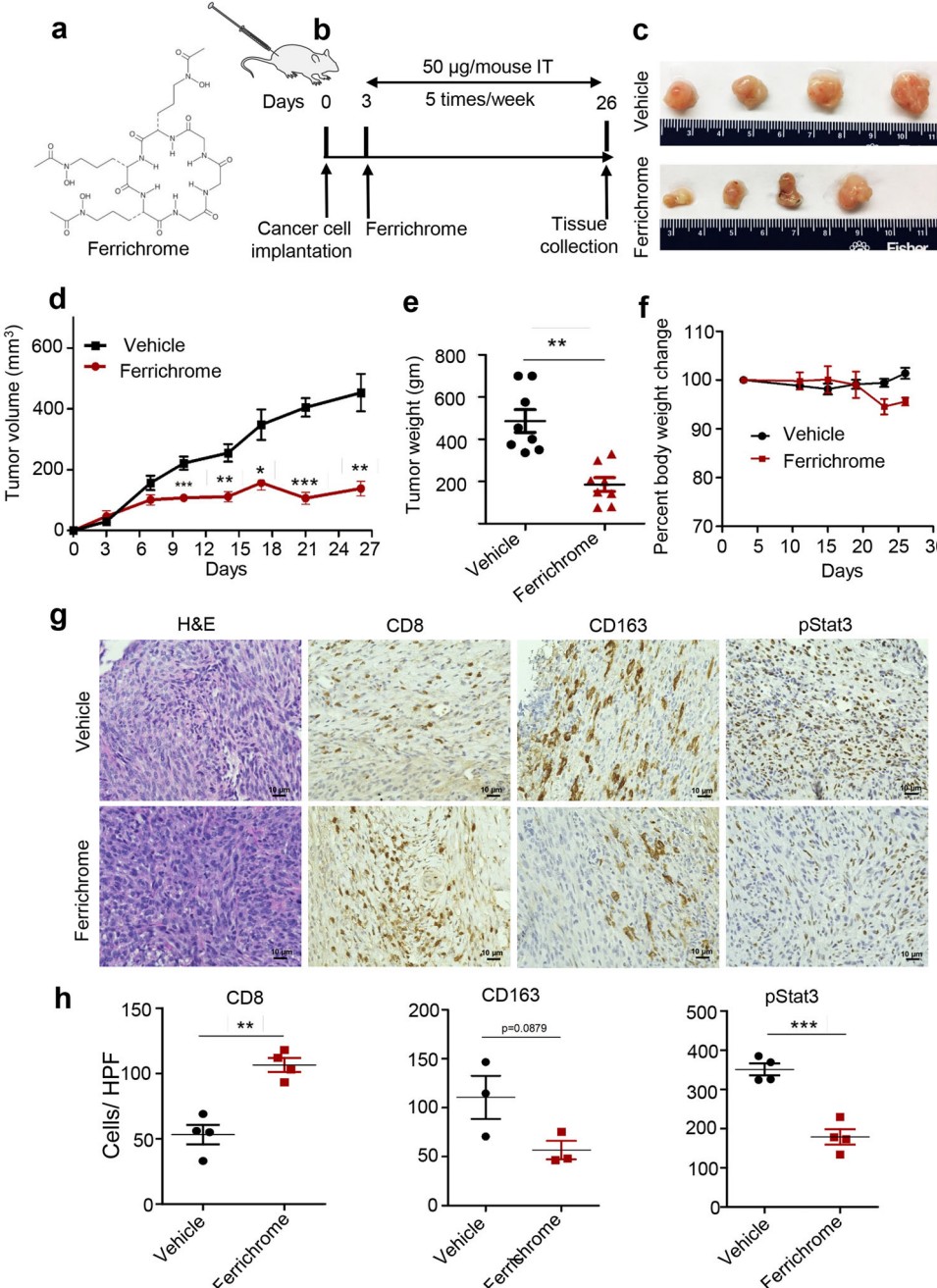

**Fig. 1 Ferrichrome suppresses tumor growth and promotes antitumor immunity in a pancreatic cancer syngeneic mouse model. a** Chemical structure of ferrichrome (iron free) (deferrichrome). **b** Schematic representation of ferrichrome treatment regimen of UN-KC-6141 cells derived syngeneic mouse model of pancreatic cancer. One million syngenetic UN-KC-6141 pancreatic cancer cells were subcutaneously injected into the right flank of C57BL/6 J mice ($n = 8$). Ferrichrome was given five times per week intratumorally (50 μM) until day 25. Vehicle consisted of equivalent volume of 1X PBS. Mice were sacrificed at day 26. **c** Representative pictures of tumors resected at day 26 from ferrichrome or vehicle-treated mice. **d** The volumes of ferrichrome- and vehicle-treated tumors ($n = 8$). **e** Wet weights of UN-KC-6141 tumors treated with ferrichrome or vehicle ($n = 8$). **f** Body weights of mice treated with ferrichrome or vehicle ($n = 5-8$). **g** Representative H&E and immunohistochemistry images of, CD8, CD163 and pStat3 staining ($n = 3-4$), Original Magnifications ×400 (**h**) Quantification of CD8, CD163 and pStat3 positive cells per high power field in ferrichrome (maroon) or vehicle-treated tumors (black) ($n = 3-4$). Mean ± SEM shown. *$p < 0.05$, **$p < 0.01$, *** and $p < 0.001$, as determined by Student's t test.

phosphorylation of signal transducer and activator of transcription 3 (pStat3) (Fig. 1g, h). STAT3 is involved in the anti-inflammatory signaling in phagocytes[42]. Taken together, these findings suggest that ferrichrome inhibits pancreatic tumor growth in a manner that is not intrinsic to the cancer cell, but through the modulation of immune milieu of the TME, potentially by decreasing macrophage-induced immunosuppression and by inducing a potent CD8 + T cell-mediated antitumor immune response.

**Ferrichrome impairs macrophage M2 polarization and promotes proinflammatory M1 phenotype.** To investigate whether the observed antitumor immune effect of ferrichrome in vivo was mediated by macrophages, mouse macrophage cell line RAW264.7 was cultured in the presence of IL-4 to induce M2 polarization, with vehicle or ferrichrome for 24 h (Fig. 2a). Gene expression analysis using quantitative real time PCR (qRT-PCR) revealed a decrease in M2 markers (*Arg1, Ym1, Mrc1, PPARγ, Fizz1*, and

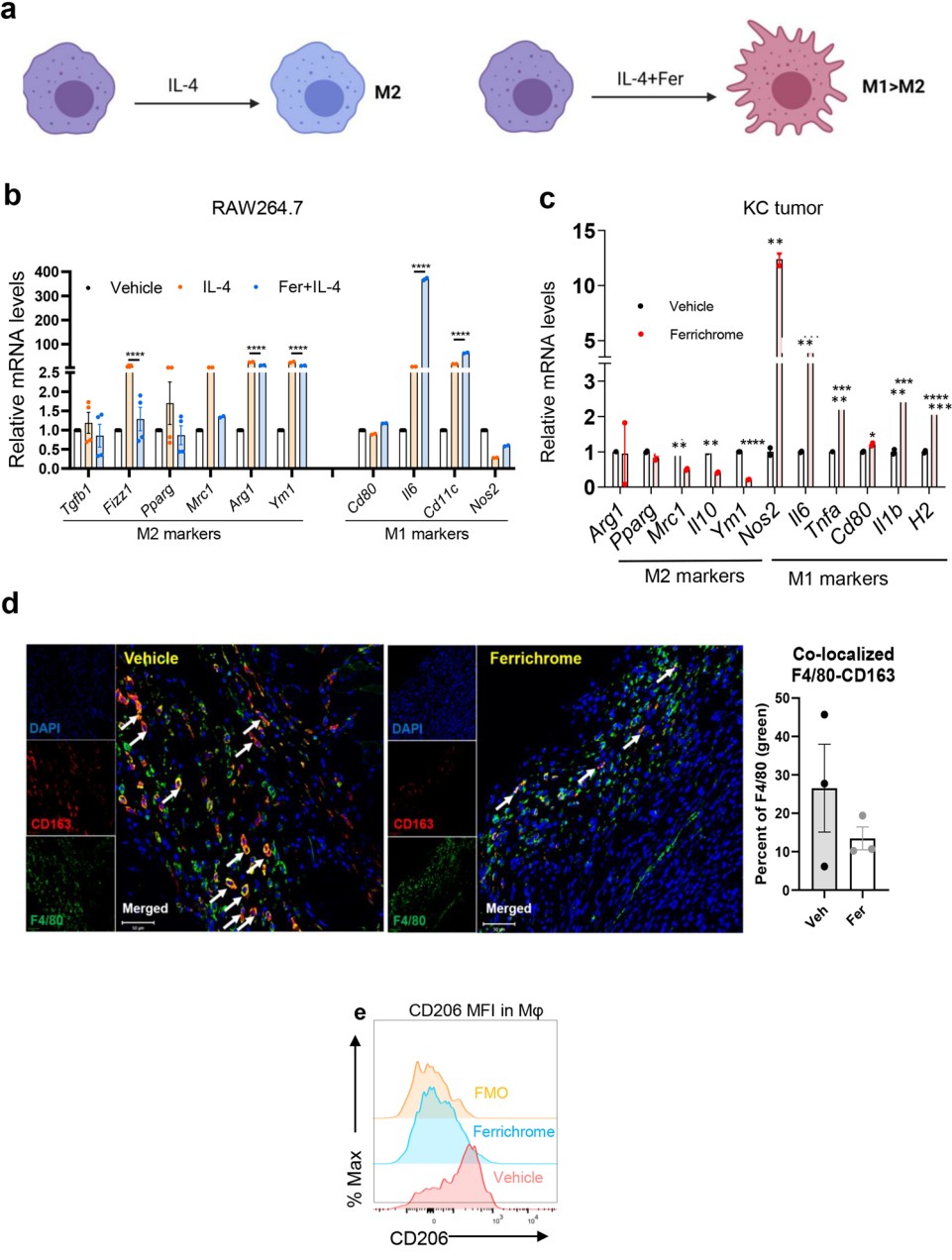

**Fig. 2 Ferrichrome abrogates M2 polarization of macrophages and promotes M1-like phenotype in vitro and in vivo. a** Representative scheme of in vitro polarization of RAW264.7 macrophages. M2 polarization of RAW264.7 cells were induced by culture with IL-4 (20 ng/mL) in the presence of ferrichrome or vehicle control (1× PBS) for 24 h. Control consisted of vehicle treated RAW264.7 cells. RNA was collected for qPCR analysis. **b** Differential gene expression of M1 and M2 biomarkers between IL-4-treated and ferrichrome (8 μM) + IL-4-treated RAW264.7 cells after 24 h of culture as evaluated by qPCR analysis. Graph is representative of two or more independent experiments with $N = 2$–4. **c** Differential gene expression of M1 and M2 markers in UN-KC-6141 tumors treated with vehicle or ferrichrome (pooled from $n = 3$ biological replicates). **d** Representative images and quantification of dual staining of macrophage markers F4/80 (green) and CD163 (red) in tumors treated with ferrichrome or vehicle as evaluated via immunofluorescence and imaged using confocal microscopy. Quantification was evaluated using ImageJ ($n = 4$). Nucleus was stained with DAPI (blue). Bar on micrographs indicates 50 μm. **e** Representative histogram of M2 marker CD206 Mean Fluorescence Intensity (MFI) in gated macrophages (CD45 + CD11b + Ly6C− Ly6G− F4/80+) of ferrichrome (blue) or vehicle-treated (red) tumor single cell suspensions as evaluated by flow cytometry analysis ($n = 4$ experiments). Fluorescence Minus One (FMO) (orange) was used as a negative control. Mean ± SEM shown **$p < 0.05$, ***$p < 0.01$, and ****$p < 0.001$, as determined by Student's $t$ test or two-way ANOVA.

*TGF-β*) and an increase in M1 markers (*Nos2*, *Il6*, *Cd11c*, and *Cd80*) in the IL-4 + ferrichrome-treated cells compared cells treated with IL-4 alone (Fig. 2b). Protein expression of M1 (iNOS) increased and M2 (Arginase 1 and CD206) markers decreased, in line with gene expression analysis of RAW264.7 cells in the ferrichrome + IL-4-treated cells as compared to IL-4-treated cells

(Supplementary Fig. 4b–c). In vivo, analysis of whole tumor RNA from UN-KC-6141 cells derived tumors revealed a similar pattern observed in in vitro experiments with increased M1 markers and decreased M2 markers in ferrichrome-treated mice compared to vehicle-treated tumors as in Fig. 1 (Fig. 2c). Importantly, total macrophage content in KC tumors was not altered by ferrichrome

(Supplementary Fig. 12C). In addition, co-staining of KC tumor sections with M2 markers CD163 and F4/80 revealed a decrease in double positive cells (M2-like macrophages) in ferrichrome-treated tumors compared to vehicle-treated tumors as assessed with dual immunofluorescence analysis, an effect that was later confirmed in other experiments (IF) (Fig. 2d). Flow cytometric analysis of single cell suspensions from tumors were consistent with IF findings showing a decrease in M2 marker CD206 mean fluorescence intensity (MFI) in macrophages (gated live CD45 + CD11b + Ly6C− Ly6G- F4/80+) in the ferrichrome-treated tumors compared to vehicle-treated tumors (Fig. 2e). These results suggest that ferrichrome has a direct effect on macrophages by inhibiting recruitment of immunosuppressive M2 TAMs while favoring a proinflammatory M1 polarization Supplementary Fig. 11d.

**Ferrichrome increases antitumor activity and phagocytic capacity in murine macrophages in vitro and in vivo.** Macrophage-mediated phagocytosis has the potential of directly killing tumor cells[43,44]. To investigate the effect of ferrichrome on macrophage phagocytic capacity, we pre-treated bone marrow-derived macrophages (BMDM) with ferrichrome for 24 h before adding *E. coli* fluorescent bioparticles for an additional 2 h of incubation. Phagocytosis was measured by quantification of bioparticles within macrophages after washing off non-ingested particles. Interestingly, ferrichrome-treated macrophages had a significantly ($p < 0.001$) higher phagocytic capacity when compared to vehicle-treated macrophages (Fig. 3a). In addition, the effect of ferrichrome on macrophage direct killing capacity was evaluated by co-culture of BMDM with luciferase-expressing PDAC mouse cell line Panc02 for 48 h in the presence of ferrichrome or vehicle treatments. Ferrichrome-treated macrophages had a significantly ($p < 0.05$) higher antitumor activity as measured by a lower tumor cell survival (decreased luciferase intensity) (Fig. 3b). However, ferrichrome did not induce apoptosis on several cancer and macrophages cell lines (Supplementary Fig. 2) and did not have a significant effect on viability of Panc02 and UN-KC-6141 cells monoculture over a large concentration range up to 150 μM but had some growth inhibitory effects (Supplementary Fig. 1a–b). We observed a moderate decrease in the proliferation of ferrichrome-treated colon cancer cell lines (SW480 and SW620) and RAW264.7 cells (Supplementary Fig. 1c–h). To verify our findings in vivo, we evaluated in vivo phagocytosis in KC tumor sections of mice treated with vehicle or ferrichrome as described in Fig. 1. Phagocytosis was evaluated using dual immunofluorescence analysis of tumor cell marker cytokeratin 19 (CK19) and macrophage marker F4/80. We observed that ferrichrome-treated tumors had a higher number of phagocytic events as compared to vehicle-treated tumors thus confirming our in vitro findings (Fig. 3c). Next, we examined key signaling pathways established to modulate macrophage phagocytic capacity. Protein death receptor (PD-1) expression by macrophages inhibits phagocytosis and antitumor immunity[45]. Moreover, macrophage phosphoinositide 3-kinase gamma (PI3Kγ, *Pik3cg*) favors an M2-like macrophage phenotype that suppresses T cell activation[46,47]. Therefore, to investigate the role of ferrichrome on macrophage gene expression of immune checkpoints PD-1, its ligand PD-L1, and PI3Kγ, as well as colony-stimulating factor receptor 1 (CSF-1R) which contributes to TAMs recruitment to the tumor site[48], we evaluated the expression of these genes *via* qPCR analysis after treatment of macrophages with ferrichrome or vehicle for 24 h. Relative gene expression analysis revealed an increase in PD-L1 and a decrease in PD-1, CSF1R and PI3Kγ levels in macrophages (Supplementary Fig. 5a–c) in response to ferrichrome treatment. However, we observed an increase in both PD-1 and PD-L1 mRNA levels in

ferrichrome-treated tumors as compared to vehicle-treated tumors (Supplementary Fig. 5d). These results suggest a role of ferrichrome in macrophage activation which subsequently results in increased phagocytic capacity and antitumor activity.

**Ferrichrome abrogates migration and invasion-promoting features of macrophages.** Factors released by M2-like TAMs promote tumor cell migration, invasion, and metastasis[49]. To investigate the role of ferrichrome on macrophage-induced migration and invasion of cancer cells, we first obtained macrophage conditioned media (CM) by culturing RAW264.7 cell line in the presence of vehicle (VCM), IL-4 (IL-4-CM) or IL-4 + ferrichrome (FCM) for 24 h. New media free of treatment and cytokines was replaced for an additional 24 h to exclude ferrichrome or IL-4 direct effect on cancer cells. Then, we treated UN-KC-6141 pancreatic cancer cells with CM for 24 h before evaluating cancer cell migration and invasion (Fig. 4a). As expected, treatment of pancreatic cancer cells with IL-4-CM significantly ($p < 0.001$) increased their migratory and invasive capacity compared to VCM (Fig. 4b, c). Ferrichrome treatment of cancer cells significantly ($P < 0.001$) reduced cancer cell migration and invasion as compared to IL4-CM treatment (Fig. 4b, c). In addition, in vivo evaluation of protein expression of matrix metalloproteinases 2 and 9 (MMP-2 and MMP-9) using IHC and their gene expression using qPCR in KC tumors revealed a decreased expression of these enzymes in ferrichrome-treated tumors as compared to vehicle-treated tumors (Fig. 4d, e). MMP-2 and MMP-9 are reported to be associated with cancer cell invasion and metastasis[50]. Also, in vitro studies showed that ferrichrome decreased the expression of *MMP12* and Stat6 genes (Supplementary Fig. 4f), which also contributes to cancer cell migration and invasion[49] and M2-like polarization[51] respectively. Taken together, these findings suggest a role of ferrichrome in abrogating macrophage-mediated promotion of pancreatic cancer cell migration and invasion.

**Ferrichrome promotes macrophage polarization toward the M1 phenotype *via* modulating TLR4-Ferroportin signaling axis.** Ferroportin is an iron export protein that is present in several cell types including macrophages[35]. Since macrophages are the major leukocyte in pancreatic tumors[9], the expression of Fpn in human samples and correlation with TAMs gene signature was examined in The Cancer Genome Atlas (TCGA) and Genotype-Tissue Expression (GTEx) databases using TIMER and GEPIA2[52,53]. SLC40A1 (Fpn gene) is highly expressed in pancreatic tumors (Fig. 5a). Surprisingly, expression of SLC40A1 is significantly higher in pancreatic tumors as compared to normal pancreas tissues in TCGA and GTEx databases, whereas HAMP expression is not differentially expressed in normal vs tumor pancreatic tissues (Fig. 5c, d). Moreover, SLC40A1 expression positively correlates with the expression of TAM genes (CD68, MRC1, IL10, CSF1, and CSF1R) in pancreatic tumors (Fig. 5b). Hepcidin, a peptide primarily expressed in the liver, is also expressed by macrophages, and plays a key role in iron metabolism by down-regulating Fpn[34,35,54]. Interestingly, previous studies have reported that hepcidin expression by macrophages is mediated by TLR4 activation and consequently results in down-regulation of Fpn and iron export by macrophages[33,36]. It is also well established that activation of TLR4 in macrophages induces their polarization to a proinflammatory M1-like phenotype[55]. Therefore, we hypothesized that ferrichrome, which is a bacterial metabolite, may induce M1 macrophage polarization by down-regulation of Fpn in a TLR4-dependent manner. To examine whether ferrichrome-induced M1-like macrophages is dependent on TLR4, RAW264.7 cells were pre-treated with the TLR4 specific

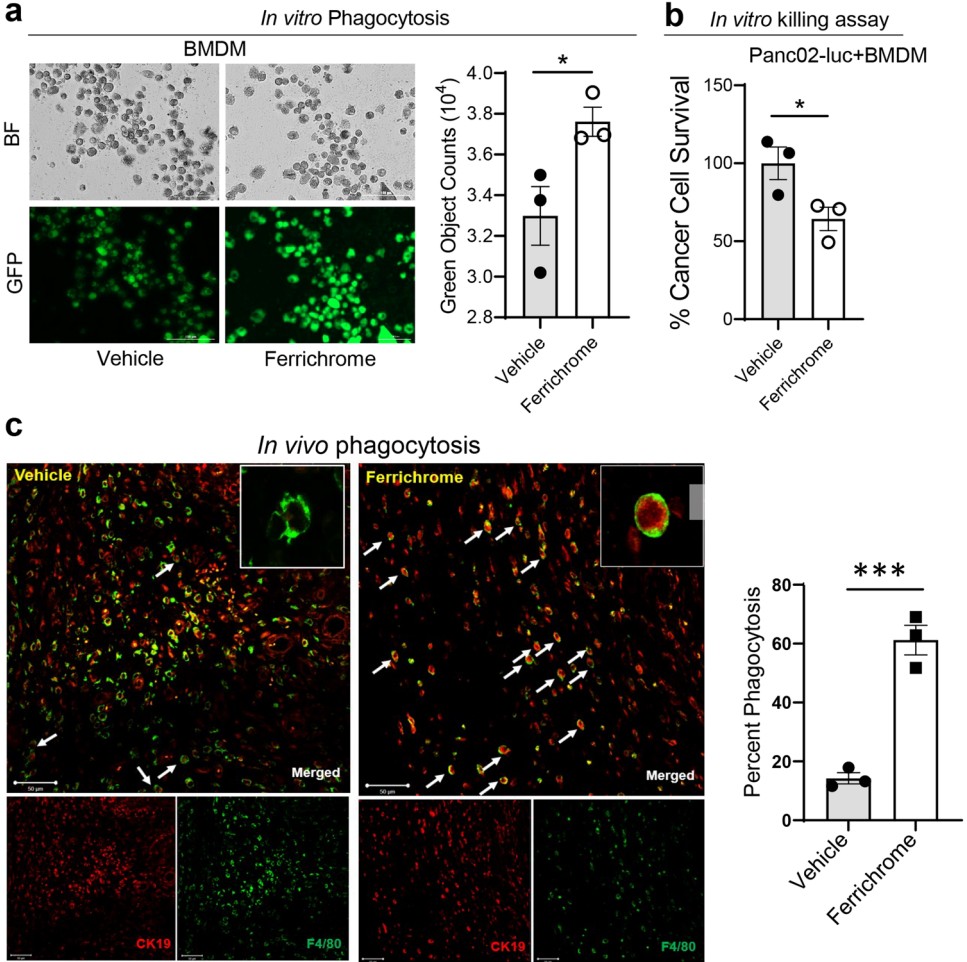

**Fig. 3 Ferrichrome promotes phagocytosis and cancer cell killing capacity in macrophages in vitro and in vivo. a** Representative image and quantification of phagocytosis of fluorescent *E. coli* bioparticles by BMDMs with the presence of ferrichrome (20 μM) or vehicle (control) as evaluated by fluorescent microscopy imaging on the FITC channel (n = 3 experiments). **b** Antitumor activity of BMDMs against luciferase-expressing pancreatic cancer cells Panc02 in the presence of ferrichrome or vehicle control. Cancer cells were co-cultured with BMDMs at a macrophage: cancer cell density of 1:1 for 48 h. Tumor cell survival was determined by normalizing luminescence to tumor-only controls (n = 3). **c** Representative images of evidence of phagocytosis of cancer cells by macrophages in ferrichrome-treated tumors (right panel) or vehicle-treated tumors (left panel). Images were quantified using ImageJ software. A phagocytic event is defined by the presence of cancer cell marker Cytokeratin 19 (CK19) (red) within the macrophage markers F4/80 (green) as evaluated by dual-immunofluorescence analysis (n = 4 mice). Bar on micrographs indicates 50 μm. Mean ± SEM shown. *p < 0.05, and ***p < 0.001 as determined by Student's t test.

inhibitor CLI-095 before vehicle or ferrichrome treatment and various proinflammatory genes (*Nos2, Tnfa, Il1b, Il6, Il12b*) were analyzed by qPCR analysis. Our results demonstrate that ferrichrome treatment alone increased the expression of *Nos2, Tnfa, Il1b, Il6, Il12b* in RAW264.7 cells while pre-treatment with CLI-095 completely abrogated ferrichrome's effect (Fig. 6g). Similar results were observed in TLR-4−/− KO derived macrophages (Fig. 6f) suggesting that M1 macrophage polarization induced by ferrichrome is dependent on the TLR4 signaling pathway. We also observed similar results using the TLR4 ligand LPS as a positive control (Supplementary Fig. 7e–h). Moreover, TLR4 inhibition by CLI-095 in RAW264.7 cells in the presence of ferrichrome increased the expression of Fpn and decreased the expression of HAMP indicating that TLR4 inhibition abrogates ferrichrome-induced inhibition of iron export phenotype in macrophages (Fig. 6h). Of note, ferrichrome treatment of RAW264.7 in the presence of iron (ferric ammonium citrate or iron sulfate, $FeSO_4$) increased gene expression of HAMP and iron storage genes Ft-L and Ft-H (light and heavy ferritin) while it decreased Fpn mRNA levels (Supplementary Fig. 6c–e). In

addition, Ft-H gene expression was also increased in RAW264.6 cells while Fpn levels were decreased even in the presence of IL-4 after treatment with ferrichrome (Supplementary Fig. 6f and a). Moreover, while treatment with either LPS or ferrichrome in the presence of iron decreased expression of Fpn in RAW264.7cells, TLR4 inhibition using CLI-095 abrogated this effect (Supplementary Fig. 7c). Next, we investigated the effect of ferrichrome on macrophage intracellular iron pool. BMDMs were treated with ferrichrome or vehicle for 48 h before the addition of iron ($FeSO_4$). Controls consisted of LPS-treated BMDMs (positive control) or no iron supply (negative control). Importantly, given the iron chelation properties of ferrichrome, cells were washed twice with PBS before the addition of iron to eliminate any effect related to iron chelation. Consistent with our Fpn and ferritin findings, ferrichrome increased intracellular iron pool in BMDMs compared to vehicle-treated BMDMs, but at a lesser extent than LPS (Supplementary Fig. 8a–b).

To further confirm our findings using genetic inhibition of TLR4, UN-KPC-960 tumor-bearing WT and TLR4−/− mice were treated with ferrichrome or vehicle control before analyzing

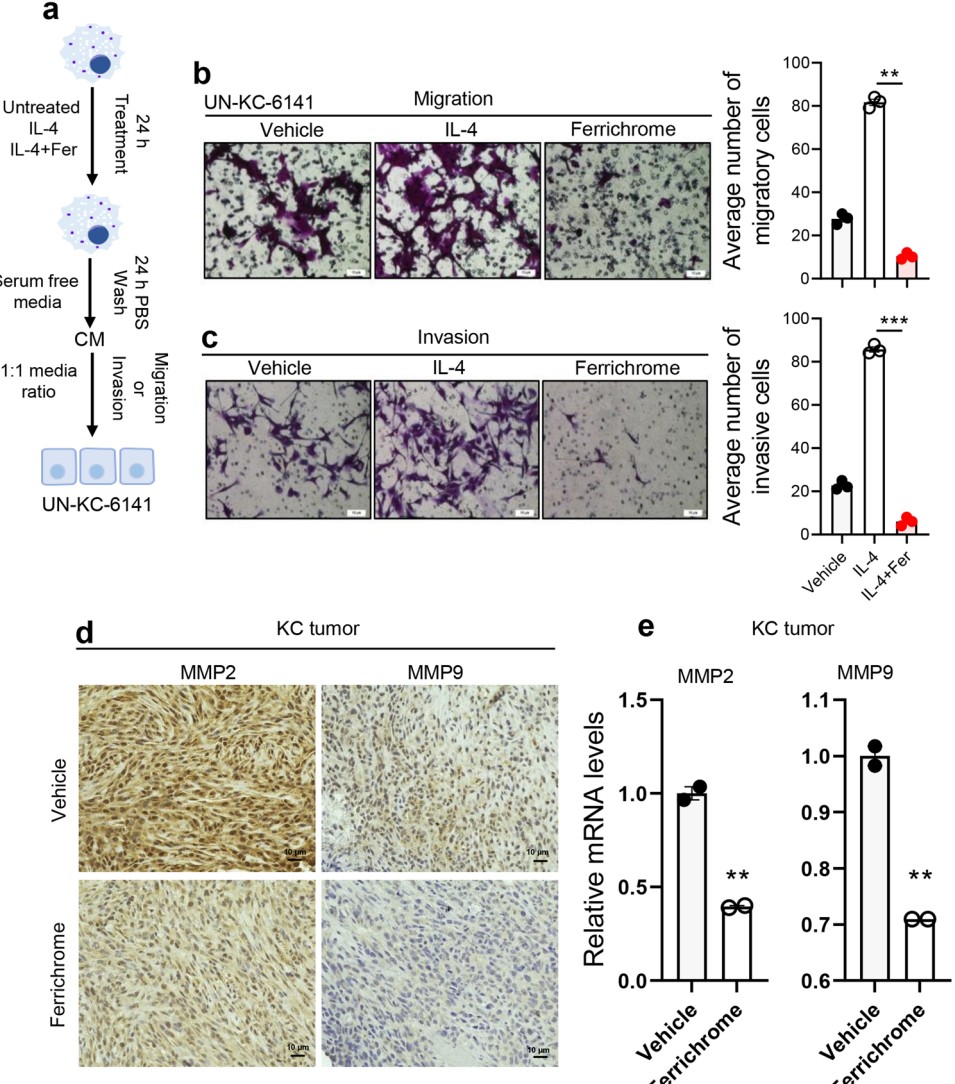

**Fig. 4 Ferrichrome abrogates macrophage-mediated cancer cell migration and invasion. a** Representative scheme of in vitro experimental design to evaluate the effect of ferrichrome on macrophage-mediated cancer cell migration and invasion. RAW264.7 macrophages were treated with vehicle, IL-4 (20 ng/mL), or ferrichrome + IL-4 for 24 h. Conditioned media (CM) was collected by culturing RAW264.7 cells for an additional 24 h in serum-free RPMI 1640 media. UN-KC-6141 pancreatic cancer cells were treated with CM at a ratio of 1:1 cancer cell media: CM for 24 H, then migration and invasion of cancer cells was evaluated via Boyden Chamber migration and invasion assays. **b** Representative images and quantification of migratory cells in # high power fields. **c** Representative image and quantification of invasive cells in # high power fields. **d** Representative immunohistochemistry images of MMP2 and MMP9 protein expression in tumors treated with either ferrichrome or vehicle ($n = 4$ mice). Original Magnifications ×400. **e** Relative gene expression analysis of MMP2 and MMP9 in tumors treated with ferrichrome or vehicle as evaluated by qPCR analysis ($n = 3$ experiments). Mean ± SEM shown. **$p < 0.01$, and ***$p < 0.001$ as determined by Student's $t$ test.

macrophage polarization in vivo *via* flow cytometry (Fig. 6a). Surprisingly, ferrichrome-treated WT mice showed a significant ($p < 0.05$) increase in MHCII$^{hi}$ M1-like macrophages and a significant ($p < 0.05$) decrease in MHCII$^{lo}$ M2-like macrophages in spleens compared to vehicle-treated WT mice (Fig. 6b, d, e). However, this effect was completely abrogated in ferrichrome-treated TLR4$^{-/-}$ mice compared to vehicle-treated TLR4$^{-/-}$ mice (Fig. 6b, d, e). In tumors, we observed a significant decrease in Arg1$^{+}$ TAMs in WT mice with ferrichrome treatment, but not in TLR4$^{-/-}$ mice (Supplementary Fig. 9a,c). We observed no significant changes in total macrophage frequencies with ferrichrome treatment in WT or TLR4$^{-/-}$ mice both in the spleen (Fig. 6c) and in the tumor (Supplementary Fig. 9b). These findings indicate that ferrichrome reverses macrophage polarization in vivo in a TLR4-dependent manner. To confirm these

findings ex vivo, murine peritoneal macrophages from wild type (WT) and TLR4$^{-/-}$ mice were treated with IL-4 to induce the M2-like phenotype in the presence or absence of ferrichrome. While ferrichrome treatment of WT macrophages increased M1 markers and decreased M2 markers as previously determined in RAW264.7 cells, this effect was abrogated in TLR4$^{-/-}$ macrophages, indicating that ferrichrome control of macrophage polarization is dependent on the TLR4 pathway (Fig. 6f). In addition, while ferrichrome treatment of WT BMDM significantly decreased *Fpn* mRNA levels, this effect was not observed when TLR4$^{-/-}$ BMDM were treated with ferrichrome (Supplementary Fig. 7b). To test whether ferrichrome also downregulates Fpn in TAMs in vivo, we co-stained KC tumor sections with Fpn and the macrophage marker F4/80. Interestingly, ferrichrome-treated tumors reduced Fpn-expressing macrophages in murine

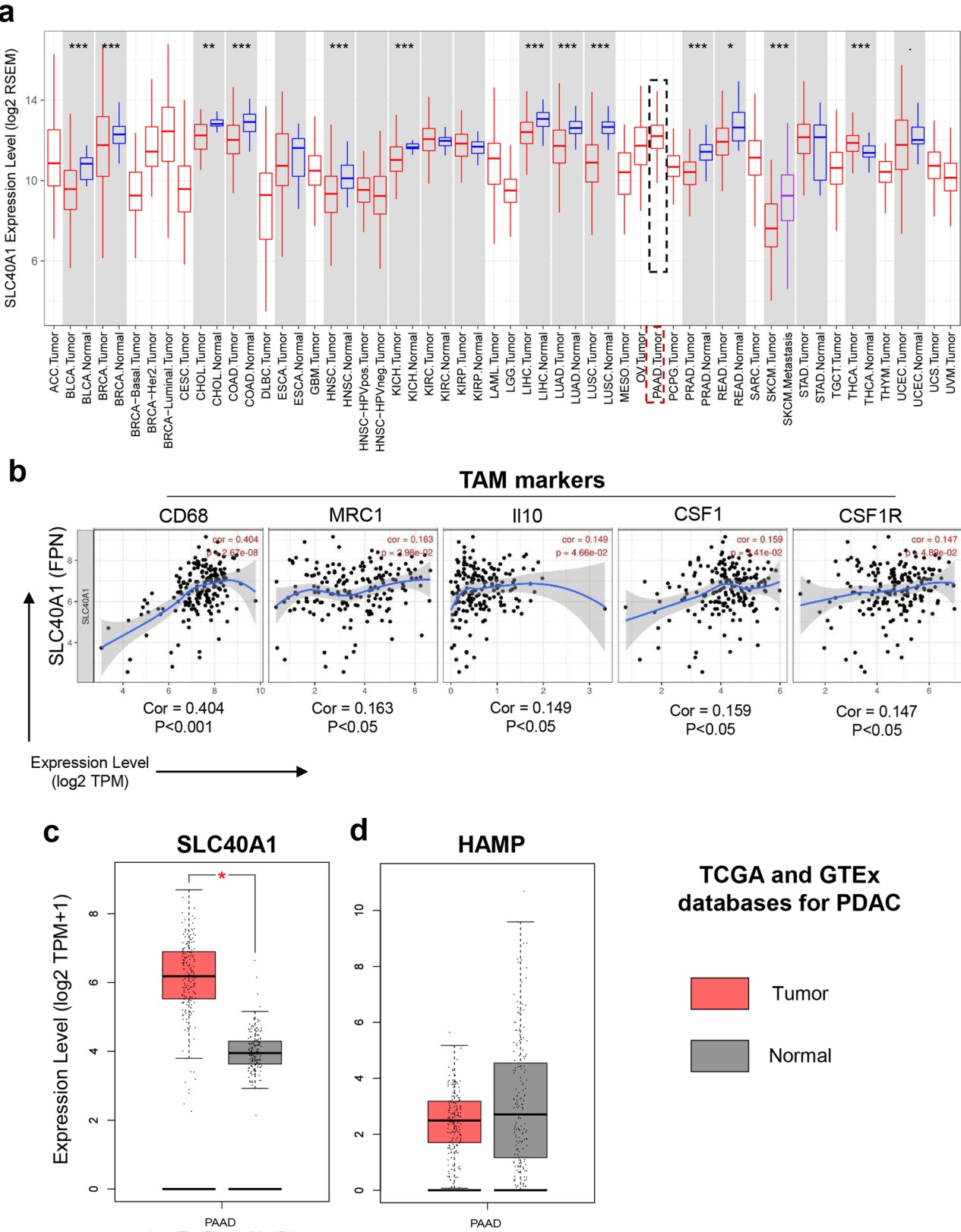

**Fig. 5 SLC40A1 expression is upregulated in pancreatic cancer tumor tissues compared to normal pancreas.** SLC40A1 also correlates positively with TAMs signature in pancreatic tumors. **a** Differential expression of SLC40A1(in blue rectangle) across different normal tissue (blue) and cancer (red) types as determined using online resource TIMER for TCGA data. **b** Correlation of SLC40A1 gene expression with TAMs gene signature in pancreatic tumors as determined using TIMER. Differential gene expression levels between normal pancreas and pancreatic tumors for (**c**) SLC40A1 and (**d**) HAMP as determined by GEPIA2 online database.

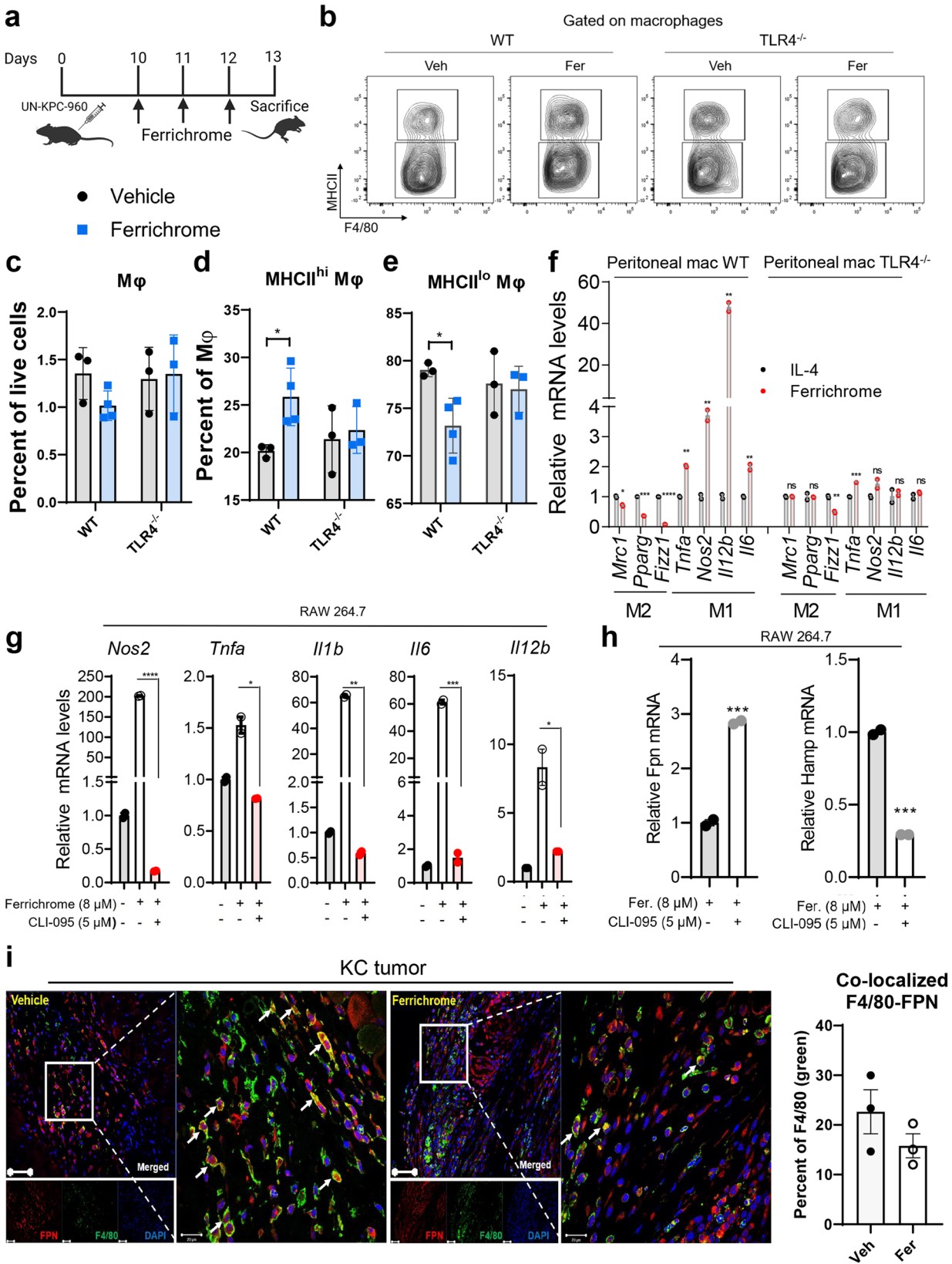

pancreatic tumors (Fig. 6i) although this effect was not significant. Because ferrichrome is a bacterial-derived iron chelator, we determined iron levels in tumors. Ferrichrome-treated tumors exhibited reduced iron levels (Supplementary Fig. 6a), yet ferrichrome did not chelate macrophage intracellular iron in vitro in RAW264.7 cells (Supplementary Fig. 6b). In addition, ferrichrome pre-treated BMDMs had higher iron retention which is a hallmark of proinflammatory macrophages (Supplementary Fig. 8a,b), increased M1 markers MHCII and CD86 (Supplementary Fig. 8c,d) and decreased M2 marker

**Fig. 6 Ferrichrome polarizes macrophages to an M1-like phenotype via down-regulation of ferroportin in a TLR4-dependent manner. a** Schematic representation of ferrichrome treatment regimen of UN-KPC-960 tumor-bearing WT and TLR4$^{-/-}$ mice. **b** Flow cytometry analysis of (**c**) total splenic macrophage content,(**d**) MHCII$^{hi}$ (M1-like) and (**e**) MHC$^{lo}$ (M2-like) splenic macrophage frequencies in WT and TLR4$^{-/-}$ mice treated with ferrichrome (blue) or vehicle control (black) ($n = 3$–4 mice per treatment group). **f** qPCR analysis of M1 and M2 markers in peritoneal macrophages isolated from wild-type or TLR4 knockout mice and treated with IL-4 alone (20 ng/mL) or ferrichrome + IL-4 for 24 h ($n = 2$–3). **g** PCR analysis of M1 markers in RAW264.7 cells treated with vehicle, ferrichrome, or ferrichrome + CLI-095 (TLR4 inhibitor) for 24 h. **h** qPCR analysis of Fpn and hepcidin mRNA levels in RAW264.7 cells pre-treated with CLI-095 in the presence or absence of ferrichrome for 24 h. Graphs are representative of at least two independent experiments with similar results. **i** Representative image and quantification of dual immunostaining of tumors treated with ferrichrome or vehicle for the macrophage marker F4/80 (green) and Fpn (red). Quantification was evaluated using ImageJ ($n = 3$ mice). Bar on micrographs indicates 50 μm. Mean ± SEM shown. **$p < 0.01$, and ***$p < 0.001$ as determined by Student's $t$ test.

CD206 (Supplementary Fig. 8e) even when BMDMs were preloaded with iron and then stimulated with either LPS or IL-4. Taken together, these findings indicate that ferrichrome polarizes macrophages to the M1-like phenotype by down-regulating Fpn levels in a TLR4-dependent manner, and not via iron chelation.

**Ferrichrome increases CD8 + T cell tumor infiltration and improves PD-L1 blockade in a murine syngeneic mouse model of PDAC.** Previous studies have shown that M2 to M1 polarization of macrophages in pancreatic tumors can orchestrate effective T cell anti-tumor immunity[56]. Therefore, we wanted to examine the effect of ferrichrome on T cell infiltration and determine whether ferrichrome can be effectively combined with ICB therapy. Interestingly, there was no significant difference in the frequency of total myeloid cell frequencies (CD11b+) or total macrophage frequencies between ferrichrome and vehicle-treated tumors was observed (Supplementary Fig. 12b–c), demonstrating that there was no defect in macrophage infiltration. However, ferrichrome-treated tumors exhibited an increased expression of MHCII on M1-like macrophages indicating that these macrophages have a mature antigen-presenting phenotype (Fig. 7a). We also observed an increase in M1-like macrophages in ferrichrome-treated tumors (Supplementary Fig. 11d). Importantly, ferrichrome-treated tumors exhibited a significant increase in total T cell and specifically CD8 + T cells as compared to vehicle-treated tumors (Fig. 7b–d). We also observed a decrease in the frequencies of both polymorphonuclear and myeloid-derived suppressor cell populations (PMN-MDSC and M-MDSC, respectively), which are known to suppress T cells, in ferrichrome-treated tumors (Supplementary Fig. 11b–c). To determine whether ferrichrome improves ICB therapy, syngeneic PDAC mice were treated with either vehicle, IgG2b isotype control, anti-PD-L1, ferrichrome, or the combination of ferrichrome and anti-PD-L1 (Fig. 7e). As in Fig. 1d, ferrichrome treatment significantly decreased tumor weight and volume (Fig. 7f–h). Anti-PD-L1 treatment alone had a marginal effect on inhibition of tumor growth. However, the presence of ferrichrome improved responsiveness to anti-PD-L1 therapy (Fig. 7f–h). There was evidence of superior clinical effect in the ferrichrome-anti-PD-L1 group compared to ferrichrome alone. Taken together, these data indicate that ferrichrome appears to improve efficacy of anti-PD-L1 treatment in a preclinical pancreatic cancer syngeneic mouse model. Collectively, ferrichrome could represent an attractive strategy to induce CD8 + T cell infiltration and improve responsiveness of ICB therapy in pancreatic cancer.

## Discussion
The influence of microbiome and probiotics has recently been established in cancer progression and cancer therapeutics[57]. The anticancer effect of certain probiotics and their metabolites have also been reported in recent studies[40]. However, the mechanisms underlying the benefit of microbiome, probiotics and their metabolites in relation to cancer progression and anticancer immunity are largely unclear[58]. Gut microbiota preferentially colonize extra-intestinal tumor sites and influence cancer therapeutics *via* STING signaling[59]. Herein, we sought to investigate the role of a microbially-derived ferrichrome in reprogramming tumor innate and adaptive immunity in preclinical in vitro and in vivo pancreatic cancer models. Our study provides new evidence that ferrichrome influences antitumor immune response in pancreatic adenocarcinoma by abrogation of M2-like polarization of TAMs while promoting M1-like polarization *via* activation of TLR4 signaling. This effect resulted to potent tumor growth inhibition by activation of macrophage-mediated antitumor innate immunity and stimulation of CD8 + T cell-mediated adaptive immunity in pancreatic tumors. Pancreatic tumors have shown very limited responsiveness to ICB therapies. Our results suggest a novel approach to improve pancreatic tumor responsiveness to ICB that might be a breakthrough in the pancreatic cancer therapy field.

Preclinical strategies for targeting macrophages increased infiltration of T cells, improved tumor immunity, and sensitized pancreatic cancer to immunotherapies[60]. These preclinical data have led to an increase in the number of clinical trials examining combination therapies to treat cancer[61]. Macrophages represent the major leukocyte population in tumors[62], thus making them an attractive target to reverse immune suppression. Therefore, we discovered a novel approach for targeting TAMs with microbially-derived ferrichrome that skews their polarization toward a proinflammatory antitumor phenotype and developed a microenvironment that facilitates immunotherapy responsiveness in pancreatic cancer. In our syngeneic preclinical pancreatic cancer mouse model, ferrichrome effectively decreased tumor burden in PDAC by increasing cytotoxic T cells population in tumors and antitumor immunity. Indeed, the role of the TME in ferrichrome-mediated efficacy was supported by the fact that treatment of mouse and human cell lines with ferrichrome in vitro did not induce apoptosis, which suggests that ferrichrome may inhibits tumor growth in vivo *via* other mechanisms such as activation of innate and adaptive immunity. However, we observed that ferrichrome partially delayed the proliferation of colon cancer cells used in the Konishi et al. study and RAW264.7 cells. Therefore, we cannot exclude that some of ferrichrome's effect on cancer cell proliferation may be *via* direct delay of proliferation. Macrophages adjust their phenotype from proinflammatory to anti-inflammatory and vice-versa in response to microenvironmental signals[55]. Moreover, macrophage phagocytosis of cancer cells plays a major role in antitumor immunity[63–66]. For instance, blockade of CD47 (which is a "do-not-eat me signal") in cancer cells synergizes with rituximab to promote phagocytosis and eradicate non-Hodgkin lymphoma[65]. In addition, anti-CD47 antibody-mediated phagocytosis of cancer cells by macrophages drives a T-cell-mediated antitumor immune response by priming CD8 T cells[66]. Our findings show that

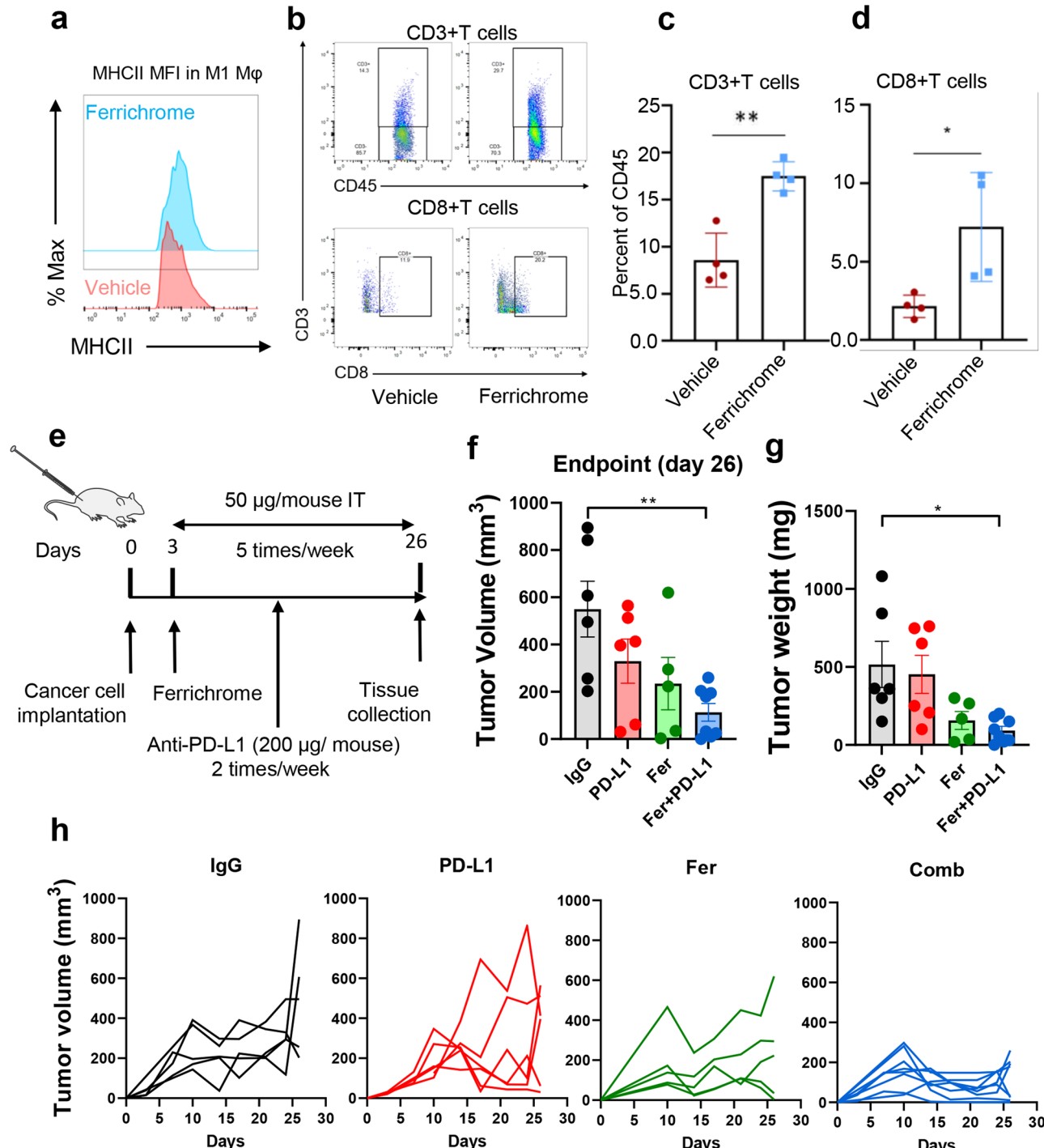

**Fig. 7 Ferrichrome promotes CD8 + T cell infiltration in pancreatic tumors and improves anti-PD-L1 therapy. a** Representative histogram comparing MHCII MFIs in M1-like macrophages (gated CD45 + CD11b + Ly6C− Ly6G- F4/80+ MHCII-high) of tumors treated with ferrichrome (cyan) or vehicle (red) as determined by flow cytometry analysis (n = 3 mice). **b** Flow cytometry analysis of the frequency of (**c**) total T cells (CD3+) and (**d**) CD8 + T cells in tumors treated with ferrichrome (blue) or vehicle (maroon) (n = 4 mice). **e** Schematic representation of combination treatment regimen of ferrichrome and anti-PD-L1 of UN-KC-6141 syngeneic mouse model. Ferrichrome was given five times per week starting at day 3 while anti-PD-L1 (200 µg/mouse) was given twice per week starting at day 5. Tumors were resected at day 26 (n = 5–8). Graph representing mean values of (**f**) tumor volume at endpoint, (**g**) tumor weight and (**h**) tumor volume of individual mice per treatment group. Mean ± SEM shown. *p < 0.05, and **p < 0.01, and ***p < 0.001 as determined by Student's t test for T cell analysis and one-way ANOVA for combination analysis.

ferrichrome promotes phagocytosis both in vitro and in vivo which also correlates with an increase in cytotoxic T cells in the tumor. Therefore, ferrichrome-mediated cancer cell phagocytosis by macrophages is an important process in antitumor immunity that may act as a dual hit by directly eliminating cancer cells, and

also by priming CD8 T cells. Furthermore, our in vitro and in vivo findings confirm a role of ferrichrome in abrogating M2-like macrophage polarization while promoting M1-like phenotype. Ferrichrome also abrogated tumor-promoting properties of macrophages such as promotion of migration and invasion,

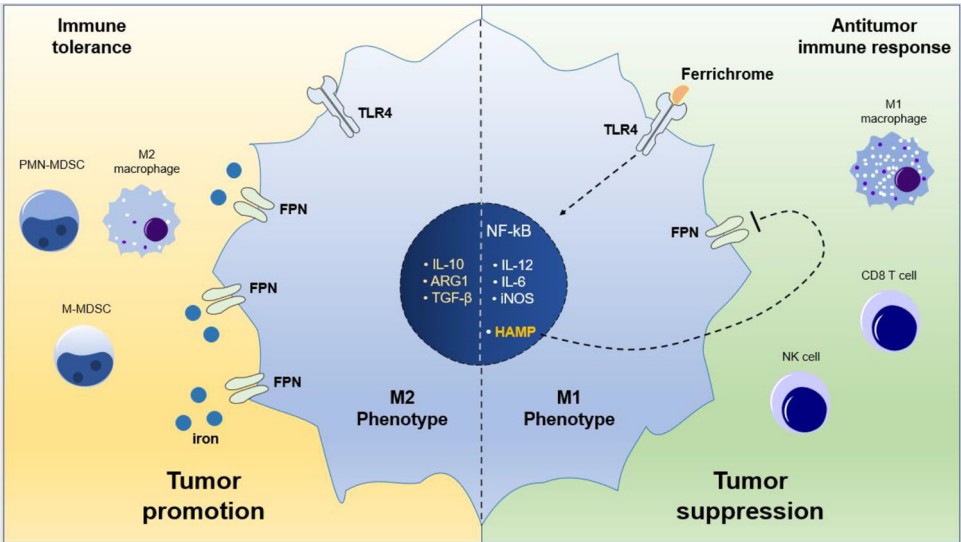

**Fig. 8 Schematic representation illustrating summarized findings.** Ferrichrome polarizes macrophages to an M1-like phenotype via TLR4 activation which results in production of proinflammatory signals. TLR4 activation by ferrichrome modulates iron metabolism via downregulation of iron export protein Fpn via endogenous production of hepcidin. Activation of M1-like macrophages drives an antitumor immune response against pancreatic tumor cells via increase infiltration of CD8 + T cell.

confirming that ferrichrome reprograms macrophage polarization from tumor-promoting to an antitumor phenotype. While our in vitro work supports that ferrichrome impacts macrophage capacity to promote cancer cell migration, invasion and aggressiveness, additional studies to determine the contribution of macrophage-specific effects are needed and planned in future studies.

Consistent with previous studies in inflammatory models linking TLR4 to Fpn[33,36], we show that ferrichrome downregulates Fpn in macrophages *via* activation of the TLR4 pathway using pharmacological and genetic loss of function in vivo and ex vivo approaches. Moreover, pharmacologic or genetic ablation of macrophage TLR4 abrogated ferrichrome-induced reversepolarization from M2-like to M1-like phenotype. Mechanistically, these findings suggest that ferrichrome modulates macrophage iron metabolism in a TLR4-dependent manner, thus inducing the macrophage polarization state to be skewed toward a proinflammatory anti-tumor phenotype.

Lastly, we aimed to determine whether ferrichrome had the potential to improve ICB therapy in a preclinical mouse model. Immunotherapy has revolutionized cancer treatment showing unprecedented long-term antitumor responses[58]. However, ICB remains largely ineffective or unresponsive in pancreatic cancer patients[67]. Our findings suggest that the presence of ferrichrome improved the responsiveness of a lead immunotherapy regimen (anti-PD-L1) while anti-PD-L1 without ferrichrome had only negligible effect on tumor growth. Previous studies have reported that PD-L1 expression on macrophages correlates positively with infiltration of CD8 T cells. Consequently, high levels of macrophage PD-L1 is associated with greater immunotherapy efficacy and longer survival in cancer patients[68]. Therefore, some of the antitumor effects observed in the ferrichrome + anti-PD-L1 combination group may be attributed to higher PD-L1 expression in the tumor, potentially through macrophage PD-L1 in particular. Finally, tumor associated macrophages promote cancer cell migration and invasion which leads to metastasis and disease severity[39]. This study provides a proof of concept that inclusion of ferrichrome might be useful for improving responsiveness and therapeutic efficacy of immunotherapies in pancreatic cancer patients.

In summary, our findings provide a novel mechanistic insight to a probiotic-derived siderophore, ferrichrome, through which it reprograms the pancreatic TME to improve tumor immune surveillance and responsiveness to immune checkpoint inhibitors (Fig. 8). This improved immune surveillance has shown potent inhibition of tumor growth via altered ratio of M1:M2 TAMs and increased CD8 + T cell infiltration in tumors. In addition, we have shown that the ferrichrome-mediated antitumor immune response also enhanced the responsiveness of murine pancreatic tumors to anti-PD-L1 treatment. Taken together, our findings demonstrate that ferrichrome has clinical potential for pancreatic cancer treatment. Additional studies are needed to investigate potential alternative mechanisms of ferrichrome's antitumor effect such as the effect on other innate immune cells, DCs and MDSCs in particular.

## Materials and methods

**Mice and tumor studies**. C57BL/6 J mice were obtained from Jackson Laboratories (000664, Bae Harbor, ME). WT and TLR4-/- mice were a kind gift of Dr. Guoyun Chen (UTHSC, Memphis, TN). Eight to twelve-week-old female mice were subcutaneously injected into the right flank with PDAC mouse cell lines UN-KC-6141 or UN-KPC-960. One million cells suspended in Dulbecco's Modified Eagle Media (DMEM) media were mixed at a ratio of 1:1 with Matrigel (BD Biosciences, San Jose, CA) and injected in a volume of 100 μL per mouse. For tumor growth studies, starting at day 3 post tumor cell injection, mice were randomly divided into two groups and treated with either 50 μg of ferrichrome (dissolved in 1× PBS) or equivalent volume of PBS intratumorally five times per week, with 2 days of rest on weekend. For combination study with anti-PD-L1, ferrichrome was administered as described above while anti-PD-L1 (BioXCell) was administered intraperitoneally at a dose of 200 μg per mouse twice per week starting at day 5. Anti-PD-L1 control consisted of equivalent volume of IgG2b isotype antibody (BioXCell, Lebanon, NH, USA). Tumor volume was measured 2–3 times per week starting at day 3 using caliper measurements. Volume was calculated as volume = length × width × height. Mice were sacrificed at day 26 and tumors were excised, weighted, processed for either flow cytometric analysis, fixed in formalin for immunohistochemistry and immunofluorescent analysis, or flash frozen in liquid nitrogen for mRNA extraction. Mice body weights were recorded twice per week to evaluate toxicity of drug treatments. For in vivo mechanistic studies, WT and TLR4$^{-/-}$ mice were injected with UN-KPC-960 cells and treated with ferrichrome as described above. Ferrichrome was administered on days 10,11 and 12 intratumorally and mice were sacrificed on day 13 for flow cytometry analysis on their spleen and tumor macrophage content. Animals were handled in accordance with protocol approval by the UTHSC Institutional Animal Care and Use Committee (UTHSC-IACUC).

**Cell lines, reagents and treatments.** Mouse pancreatic cancer cells lines UN-KC-6141 and luciferase-expressing Panc02 were a kind gift from Dr. Batra's laboratory at University of Nebraska Medical Center and Dr. Kazuaki Takabe's laboratory at the Roswell Park Cancer Institute in Buffalo, NY, respectively. UN-KPC-960 were a kind gift of Dr. Evan Glazer at UTHSC, Memphis TN. UN-KC-6141 cells were cultured in complete DMEM medium while luc-Panc02 cells were cultured in complete RPMI-1640 medium. UN-KPC-960 were cultured in complete DMEM-F12 medium. SW620 human colon cancer cell line was a kind gift from Dr. David Shibata at UTHSC, Memphis TN. SW620 cells were cultured in complete RPMI medium. SW480 cells were obtained from ATCC and grown in DMEM complete media containing 10% fetal bovine serum, and 1% antibiotic and antimycotic solution. RAW264.7 murine macrophage cell line was obtained from American Type Culture Collection (ATCC, TIB-71, Manassas VA) and cultured in complete RPMI-1640 medium. Iron free ferrichrome was purchased from Sigma (purity > 98%) was dissolved in 1X PBS and in vitro treatments consisted of 8 μM concentration unless specified otherwise. Endotoxin contamination was checked using the LPS detection kit (GeneScript, Piscataway, NJ, USA) to account for any potential TLR4 ligand contamination (such as LPS) that may alter interpretation of findings. Interleukin 4 (IL-4) (Sigma, St. Louis, MO) was used at a concentration of 20 ng/mL and lipopolysaccharide (LPS, L2880, Sigma) was used at a concentration of 100 ng/mL. Both were dissolved in 1X PBS. CLI-095 (Supplementary Table 2) was dissolved in dimethyl sulfoxide (DMSO) and used as a pre-treatment at a concentration of 5 μM. Ferric ammonium citrate (Sigma, St. Louis, MO, USA) and ferrous sulfate (FeSO4) (Sigma) were both dissolved in 1X PBS and used at a concentration of 100 μM. For the complete list of reagents, please refer to Supplementary Table 3.

**RNA preparations, quantitative real time PCR and Western immunoblot analysis.** RNA was extracted from cell lines, primary cells, and tumor tissues using TRIzol (Invitrogen, Carlsbad, CA, USA). Quantitative real time polymerase chain reaction (qRT-PCR) was performed as previously described[69]. For co-culture experiment, RAW264.7 cells were plated on a 0.4 μm transwell chamber while UN-KC-6141 cells were plated in the bottom well at a density of 500,000 cells per well for each cell type. Macrophages and cancer cells were cocultured with the presence of ferrichrome or vehicle for 48 h, then macrophages on the upper chamber were collected for mRNA isolation qRT-PCR analysis. Gene expression was normalized to 18 S rRNA. Primer sequences were used from Harvard Primer Bank[70], and from the following sources[33,71]. Primer sequences are listed in Supplementary Table 3. Immunoblot analysis on RAW264.7 cells and primary macrophage cells was conducted as previously described[69].

**In vitro phagocytosis and killing assays.** In vitro phagocytosis assay was performed using Vybrant Phagocytosis Assay Kit (ThermoFisher Scientific, Waltham, MA, USA) according to the manufacturer's protocol. Briefly, BMDM plated at a density of 100,000 cells per well in a 96 well plate (Corning, Tewksbury, MA, USA) in complete RPMI-1640 medium were pre-treated with ferrichrome overnight then medium was replaced by 100 μL of fluorescent BioParticle suspension containing fluorescent *E. coli* bioparticles. Cells were then incubated at 37 °C 5% $CO_2$ for 2 h, washed twice with 1X PBS to remove non-phagocytosed particles, resuspended in 1X PBS and fluorescence was analyzed using FITC channel on Cytation 3 Cell Imaging (Biotek, Winooski, VT, USA). Phagocytosis was quantified by evaluating total number of Fluorescent Green Object Count. For killing assay, luciferase expressing Panc02 mouse pancreatic cancer cells were incubated with BMDM in the presence of ferrichrome or vehicle for 48 h at 37 °C and 5% $CO_2$. Cancer cells viability was evaluated by measuring luciferase intensity as described[64] using Cytation 3.

**Cell migration and invasion.** RAW264.7 macrophages were treated with vehicle, IL-4, or IL-4 + ferrichrome for 24H. Then medium was removed, and cells were washed twice with 1X PBS to remove all traces of treatments, then medium was replaced with fresh serum-free RPMI-1640 for another 24 h. Conditioned medium (CM) was collected by spinning down culture medium at 2000 RPM and was then stored at −80 °C for further experiments. Migration and invasion experiments were done by treating mouse cancer cell line UN-KC-6141 with different RAW264.7 CM at a ratio of 1:1 CM: cancer cell media. Migration and invasion experiments were conducted as previously described[72,73].

**Cell viability and apoptosis assays.** The effect of ferrichrome on mouse pancreatic cancer cell lines UN-KC-6141 and luc-Panc02, human colon cancer cell lines SW620 and SW640 and murine macrophages (RAW264.7 and BMDMs) proliferation for the indicated time points was determined using the 3-(4,5-dimethylthiazolium- 2-yl)-2, 5-biphenyl tetrazolium bromide (MTT) assay as previously reported[69]. Apoptosis was evaluated on SW620, UN-KPC-960 and BMDMs using the FITC Annexin V kit (Biolegend, San Diego, CA, USA) according to manufacturer protocol.

**Iron staining and quantification.** Iron staining was performed on paraffin embedded tumor sections, RAW264.7 cells and BMDMs using the Iron Stain kit (Abcam, Cambridge, UK) according to manufacturer's protocol with minor modifications. Briefly, tumor sections were deparaffinized, hydrated and incubated

in a 1:1 solution of Potassium Ferrocyanide and Hydrochloric Acid Solution for 25 min. After rinsing the slides thoroughly with distilled water, Nuclear Fast Red solution was added for 5 min. After rinsing, slides were dehydrated in 95% alcohol followed by absolute alcohol, then mounted in synthetic resin. Images were taken using a bright field microscope. Iron quantification was determined by measuring optical density at 630 nm as previously described (PMID: 19998319) using Cytation 5 Multi-Mode Reader (Biotek, Winooski, VT, USA).

**TIMER and GEPIA2 databases analysis.** TIMER2.0 is a comprehensive resource for analysis of immune cell infiltration across diverse cancer types (https://cistrome.shinyapps.io/timer/)[52]. Differential analysis of SLC40A1 across different types of cancer and correlation of SLC40A1 expression with different TAM-associated genes in PDAC patients was done as previously described[74]. The online database Gene Expression Profiling Interactive Analysis (GEPIA) (http://gepia.cancer-pku.cn/index.html) was used to investigate differential expression of SLC40A1 in normal pancreas vs PDAC tumors[75].

**Immunofluorescence and immunohistochemistry analysis.** Dual immunofluorescence and immunohistochemistry analysis was performed as previously described[76]. Dual IF quantification was done using ImageJ to analyze co-localization. For IHC quantification of CD8, CD163, and P-STAT3 positive cells, three different 400X fields were used and averaged per each biological replicate ($n = 3$–4 biological replicates). For the list of antibodies used in these experiments, please refer to Supplementary Table 2.

**Flow cytometric analysis.** Tumor single cell suspensions were obtained using Mouse Tumor Dissociation Kit (Miltenyi Biotec, Auburn, CA, USA). Spleen single cell suspension were obtained by crushing spleen tissues against a 70 μm cell strainer using a syringe plunger. Single cell suspensions were incubated with Ghost dye (Tonbo, San Diego, CA, USA) to assess cell viability, FcR-blocking reagent (Tonbo) and fluorescently labeled antibodies and incubated for 30 min as previously described[77]. Controls consisted of single color Ultracomp Beads (Invitrogen, Carlsbad, CA, USA) positive and Fluorescence Minus One (FMO) negative control. Data were acquired using high-performance Bio-Rad ZE5 flow cytometer (Bio-Rad, location) and analyzed using the flow cytometry analysis program FlowJo (Tree Star, Ashland, OR, USA). Gating strategy for immune cell populations is detailed in (Fig. S11). For complete list of flow cytometry antibodies and reagents used in this manuscript, please refer to Supplementary table S1.

**Statistics and reproducibility.** Statistical analyses were performed using GraphPad Prism software by using Student's *t* test when comparing two variables or two-way ANOVA when comparing more than two variables. Data are expressed as mean ± standard error (SE) values. We considered statistically significant when calculated differences were *$p < 0.05$, **$p < 0.01$, and ***$p < 0.001$. Reproducibility of our study was ascertained by performing multiple experimental sets and data analyses through two different statistical tests.

**Reporting summary.** Further information on research design is available in the Nature Research Reporting Summary linked to this article.

## Data availability

Uncropped Western blot images were provided in supplementary Fig. 12D. All relevant data including the numerical and statistical source data used in main graphs and figures were provided in Supplementary Data 1–6.

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

## Acknowledgements

National Institute of Health grants R01 CA210192, R01 CA206069, R01 CA204552 awarded to Dr. Subhash C. Chauhan and NIH SC1 GM140982 grant awarded to Dr. Bilal B. Hafeez. We give special thanks to Kosten Foundation for funding part of this project. The Molecular Resource Center (fluorescence measurements) and Flow cytometry facility of University of Tennessee Health Science Center for help in acquisition of data is highly acknowledged. The Mary Kay Foundation and NIH R01CA253329 to Dr. Liza Makowski. We thank T.J. Hollingworth for his excellent technical help in confocal microscopy, Chidi Zacheaus assisting with mouse models, and Dr. Guoyun Chen for providing TLR4$^{-/-}$ and control mice. We thank Dr. Evan Glazer for providing us with UN-KPC-960 cells and Dr. David Shibata for providing us with SW620 cells.

## Author contributions

M.C., B.B.H., and S.C.C. conceived the idea and designed the project. M.C. and B.B.H. carried out most of the experiments and analyzed the initial data. H.M., S.Kumari, D.D., A.K.P., and E.A. helped in different experimental procedures. M.C., M.S., V.K.K., and B.B.H. performed xenograft and immunotherapy experiments and GYC provided TLR-/- mice. L.M., S.B., M.J., S.K., M.K.T., and M.M.Y. helped in data review, discussion, shared laboratory resources and provided meaningful scientific insights in the project. M.C., B.B.H., and S.C.C. compiled the figures. M.C., B.B.H., L.M., and S.C.C. wrote, edited, and revised the paper. S.C.C. provided majority of laboratory resources and edited the final paper.

## Competing interests

The authors declare no competing interests.
