## [Peer Review File · Communications Biology]

Reviewers' comments:

Reviewer #1 (Remarks to the Author):

This manuscript investigates how ferrichrome, a bacterially derived iron chelator, can impact macrophage differentiation and promote anti-tumor effects in pancreatic cancer. Major conclusions are that ferrichrome acts independent of iron-binding to activate TLR4, induce M2 to M1 macrophage skewing, and enhance anti-tumor immunity. This area of investigation is of high importance, as checkpoint inhibitors are known to be ineffective against many cancers and in many patients. Enhancing cancer immunotherapy and understanding the mechanisms behind this will improve the care of many cancer patients.

Anti-tumor effects of ferrichrome in pancreatic cancer models are quite clear. Other data that explore mechanism are potentially exciting, but remain preliminary and require additional investigation to define what is the primary mechanism driving the anti-tumor effects of ferrichrome treatment.

One major concern is that all observations with ferrichrome could be a result of LPS contamination. The authors must demonstrate that the effects of ferrichrome are independent of other TLR4 ligands. Konishi, Nat Comm 2016 (ref 40) uses several ways to precipitate out ferrichrome, and LPS detection kits are commercially available. In addition, the role of iron is not sufficiently explored.

Additional critiques:

The authors conclude from Figure 1 and S1 that "ferrichrome inhibits pancreatic tumor growth in a manner that is not intrinsic to the cancer cell," as viability is minimally affected in their pancreatic cancer cell lines. However, Konishi, Nat Comm 2016 (ref 40) have shown that ferrichrome impacts cell proliferation/death of Caco2 and SW620 colorectal cancer cells. This current experiment requires a positive control, such as Caco2 and/or SW620, to demonstrate that pancreatic cancer cells are unique and do not respond to ferrichrome treatment. In addition, as ferrichrome is reported to induce apoptosis, it is important to specifically evaluate apoptosis in ferrichrome-treated cancer cells and macrophages.

M1/M2 evaluation: While the data in Fig 2b are easy to follow because the baseline (IL-4 stim) is set to 1, the authors should in fact normalize their data to an untreated macrophage. This is essential as macrophage cell lines like Raw 264.7 are difficult to differentiate to M2, thus a control is needed to show that M2 polarization is occurring with IL-4. In figure 2c, the authors should assess a general macrophage marker to show whether macrophage abundance is altered.

Conclusions from 2D are rather strong, given that only a representative image is shown. As a result, they do not support the conclusions (this is also a problem with fig 3d and 6d) and an additional method of analysis is required. The authors can clearly use flow cytometry to analyze these macrophages (Fig 2e), which is a possible approach they could take to generate quantitative co-localization data.

What is the relative importance of macrophage-mediated phagocytosis of cancer cells, vs. other mechanisms including NK or CD8 T cell mediated killing? From my knowledge, macrophage-mediated phagocytosis is not a primary means of anti-tumor immunity. Does blocking macrophage-mediated phagocytosis lead to significant tumor progression?

The sections on PD-L1, phagocytosis and killing, as well as cancer cell invasion and migration, are mainly correlative. It's unclear how these observations explain the in vivo results that ferrichrome prevents tumor growth in the subcutaneous model. Furthermore, it is difficult to envision how these various pathways and macrophage activities may be related and contribute to anti-tumor effects.

Figure 7 begins to explore an anti-tumor mechanism involving CD8 T cells, but goes no further than

showing T cell numbers and reduced tumor volume when Ferrichrome is combined with anti-PD-L1. I'm unsure what this figure contributes to our mechanistic understanding of anti-tumor immunity by ferrichrome. Importantly, anti-PD-L1 does not significantly reduce tumor volume, and the reduction by ferrichrome plus anti-PD-L1 does not appear different from ferrichrome alone (Fig 1). Thus ferrichrome does not increase the efficacy of anti-PD-L1 treatment, as claimed.

Minor:

Fig 2a is confusing to me – if M2 markers are reduced, why does the end of the pathway indicate M2>M1?

Fig 2b: TGF-B is not an M1 marker

Fig 2 and others: Why are different M1/M2 markers used for human vs. mouse?

Reviewer #2 (Remarks to the Author):

The current study by Chaib et al investigated the anti-tumorigenic effects of ferrichrome (*Lactobacillus casei* derived siderophore) on a murine pancreatic cancer model. Specifically, intratumoral administration of ferrichrome reprogrammed tumor-associated macrophages (TAMs), increased CD8+ T cell infiltration into tumors and markedly reduced tumor burden. The authors made an interesting observation that altered immune response by ferrichrome improved anti-PD-L1 therapy. The authors also show that ferrichrome can induce TAMs polarization via activation of the TLR4 pathway that represses the expression of iron export protein ferroportin (FPN1) in macrophages. The results are intriguing, however there are some major concerns which should be addressed.

Major comments:

1. Although the data suggests that ferrichrome is able to activate TLR4, no direct evidence has been provided. Authors need to demonstrate ferrichrome's ability to induce classical pro-inflammatory cytokines of TLR4 pathway in vivo using WT and TLR4-deficient mice. Alternatively, authors could use the commercially available reporter cell lines, like TLR4-expressing HEK293 cells, to substantiate ferrichrome-TLR4 interaction.
2. The source of ferrichrome is not provided. What is the purity of ferrichrome? Any endotoxin contamination? Is it iron free? Most importantly, authors need to perform some of the critical in vitro experiments in the presence of Polymyxin B. These need to be rigorously addressed.
3. It is not clear whether the iron chelation property of ferrichrome exert effects on the tumor model. Authors stated that: "Moreover, we elucidated that ferrichrome regulates macrophage iron metabolism by decreasing Fpn expression in a TLR4-dependent manner, not by iron chelation" (lines # 113-115). However, the authors measured ferritin as an indirect marker, but did not use methodology that directly assess changes in the intracellular iron levels in ferrichrome treated cells.
4. Is this anti-tumoral property is unique to ferrichrome? Or any other water soluble siderophores (e.g. DFO) can do? Addressing this point can provide insight on whether the anti-tumor effects are independent of iron chelation.

Minor comment:

5. The authors did not show anti-tumorigenic effects of *Lactobacillus casei*. Whether *L. casei* (heat killed or live) in combination would improve the efficacy anti-PD-L1?
6. Supplementary Figure: 2A, lower panel marked wrongly, it should be M1>M2.

Dear Dr. Moroishi,

We have addressed all the reviewers' comments. Below is our point-by-point response of all the reviewers' comments. Our **responses are below in red**. Changes were made are in yellow highlighted in revised manuscript.

Comments of Reviewer #1:

Comment # 1: *This manuscript investigates how ferrichrome, a bacterially derived iron chelator, can impact macrophage differentiation and promote anti-tumor effects in pancreatic cancer. Major conclusions are that ferrichrome acts independent of iron-binding to activate TLR4, induce M2 to M1 macrophage skewing, and enhance anti-tumor immunity. This area of investigation is of high importance, as checkpoint inhibitors are known to be ineffective against many cancers and in many patients. Enhancing cancer immunotherapy and understanding the mechanisms behind this will improve the care of many cancer patients.*

Response: We thank reviewer for the careful consideration of our manuscript. We feel that addressing your concerns have really made the manuscript much stronger.

Comment # 2: *Anti-tumor effects of ferrichrome in pancreatic cancer models are quite clear. Other data that explore mechanism are potentially exciting but remain preliminary and require additional investigation to define what is the primary mechanism driving the anti-tumor effects of ferrichrome treatment. One major concern is that all observations with ferrichrome could be a result of LPS contamination. The authors must demonstrate that the effects of ferrichrome are independent of other TLR4 ligands. Konishi, Nat Comm 2016 (ref 40) uses several ways to precipitate out ferrichrome, and LPS detection kits are commercially available.*

Response: We thank reviewer for pointing out this critical issue. We agree that confirming endotoxin contamination in our Ferrichrome stock is a paramount concern. Unfortunately, Sigma confirmed that they don't measure endotoxin levels in ferrichrome batches. Therefore, using an LPS detection kit GeneScript L00350C, we tested the Ferrichrome batch used in this manuscript. Endotoxin concentrations in control standard curve ranged from 0.01- 0.1 EU/ml standard and ferrichrome measured below the lowest concentration at 0.017 EU/ml. These new data now demonstrate that endotoxin is below 0.01 concentration therefore not likely to mediate impacts observed herein. We have added these results in supplemental **Fig. 3** and described in results and methods section of revised manuscript.

Comment # 3: *In addition, the role of iron is not sufficiently explored*

Response: We explored intracellular iron levels in macrophages as described in experiments below in response to **reviewer #2 question 3**.

Comment # 4: *The authors conclude from Figure 1 and S1 that "ferrichrome inhibits pancreatic tumor growth in a manner that is not intrinsic to the cancer cell," as viability is minimally affected in their pancreatic cancer cell lines. However, Konishi, Nat Comm 2016 (ref 40) have shown that*

ferrichrome impacts cell proliferation/death of Caco2 and SW620 colorectal cancer cells. This current experiment requires a positive control, such as Caco2 and/or SW620, to demonstrate that pancreatic cancer cells are unique and do not respond to ferrichrome treatment.

Response: It is an important point. Not every cancer cell type may be susceptible. As you rightly point out, other cells have been reported to be susceptible to ferrichrome-induced death. To address your concerns, we also examined the effect of Ferrichrome on cell proliferation/death by MTT assay at 2 time points using 5 doses in 2 colon cancer cell lines (SW480 and SW620). We observed that ferrichrome had moderate inhibitory effect in both SW480 and SW620 colon cancer cells at 48h time point, but no significant inhibitory effect at 24h time point in SW480. There was moderate dose dependent inhibition of proliferation in SW620 (new Supplementary Fig. 1 C-F). Therefore, this positive control shows that ferrichrome reduces cell numbers in SW620 in support of Konishi et al. However, we did not observe an increase in apoptosis with ferrichrome treatment of SW620 cells as reported by Konishi et al (Supplementary Fig. 2C). Furthermore, in support of our work, another murine pancreatic cell line UN-KPC-960 and murine macrophages (RAW264.7) for MTT in SF1 and BMDMs by flow in SF2A), ferrichrome did not inhibit cell proliferation or induce apoptosis. Therefore, in both mouse and human pancreatic cell lines, we have demonstrated that ferrichrome does not dramatically impact on cell proliferation or induce apoptosis while colorectal cancer cells show moderate inhibition of proliferation. We have included this data as new supplemental figure SF1C-H and noted details of experiment in methods, results, and in discussion, we note this difference between our findings and those of Konishi et al. in apoptosis.

Comment # 5: *In addition, as ferrichrome is reported to induce apoptosis, it is important to specifically evaluate apoptosis in ferrichrome-treated cancer cells and macrophages.*

Response: To address reviewer concerns, we measured apoptosis by flow cytometry for annexin V on murine pancreatic cancer cells UN-KPC-960 (KPC cells), BMDM and SW620 (Supplementary Fig. 2). As we discussed above, ferrichrome did not induce apoptosis in these cells across a dose range from 10- 1000 ng/ml.

Comment # 6: *M1/M2 evaluation: While the data in Fig 2b are easy to follow because the baseline (IL-4 stim) is set to 1, the authors should in fact normalize their data to an untreated macrophage. This is essential as macrophage cell lines like Raw 264.7 are difficult to differentiate to M2, thus a control is needed to show that M2 polarization is occurring with IL-4.*

Response: We agree with the reviewer's important points. We have re-graphed to incorporate fold changes of M1 and M2 markers in IL-4-treated Raw264.7 cells by normalizing to untreated (vehicle) control in supplementary figure 4 as you suggest. Compared to vehicle treated, 4 of the 6 M2 markers measured were significantly 2-25-fold elevated (*Fizz1*, *Mrc1*, *Arg1*, *Ym1*) demonstrating that despite RAW264.7 macrophages being primarily M1-like, IL-4 treatment significantly increased M2 markers. New supplemental Fig 4 layout is now shown.

Comment # 7: *In figure 2c, the authors should assess a general macrophage marker to show whether macrophage abundance is altered.*

Response: We agree and realized that we were not clearer regarding macrophage abundance in tumors. In fact, we did measure total macrophage abundance in tumors by flow cytometry. Macrophage frequencies showed no significant alteration in macrophage content as analyzed by flow cytometry using the following gating for macrophages: CD11b+ Ly6C- Ly6G- F4/80+. This data was shown in supplementary Figure 13C. To address your concerns, we more clearly discussed this finding by stating “total macrophage content in KC tumors was not altered by ferrichrome treatment (Supplementary Fig. 13C)” on lines 176, 177.

Comment # 8: *Conclusions from 2D are rather strong, given that only a representative image is shown. As a result, they do not support the conclusions (this is also a problem with fig 3d and 6d) and an additional method of analysis is required. The authors can clearly use flow cytometry to analyze these macrophages (Fig 2e), which is a possible approach they could take to generate quantitative co-localization data.*

Response: We thank reviewer 1 for this important comment about Fig 2D overstating the finding that ferrichrome reduced M2 content in the TME based on colocalized IF imaging by showing a representative image of DAPI/CD163/F4/80. To address your concerns, quantification using ImageJ in figures 2D has been incorporated. Fig. 2D now quantifies immunofluorescence for M2-like macrophages using the M2 marker CD163 shown in new graph in Fig 2D to the right of IF images. While the difference is not significant due to high variance, the Ferrichrome treated KC tumor have about 2-fold reduced M2 macrophages. To further address the rigor for M2 quantification, additional M2 markers including surface marker CD206 quantified in new Fig. 2E by flow and intracellular enzyme arginase (Arg1) quantified in new figure Sup Fig. 10 by flow have also been included to support Ferrichrome mediated downregulation of M2 macrophages. Therefore, using three complementary markers and methods, we report that Ferrichrome downregulates M2 macrophages in the TME.

To address the rigor noted for Fig 3d, wherein we examined “in vivo” phagocytosis shown by immunofluorescence of epithelial cancer cell label CK19 and F4/80 as previously reported by Liu et al (PMID: 30664738) as a biomarker of in vivo phagocytosis by examining engulfment of CK19+ cells by F4/80+ cells. While there is no other way to establish in vivo phagocytosis, we conducted two assays to support our work shown in Fig. 3A-C we include supportive evidence including two in vitro studies. Fig 3A demonstrates that Ferrichrome significantly induced engulfment of E.Coli beads by quantification of IF images in Fig 3b. Fig 3c also shows impacts of ferrichrome to significantly reduce cancer cell survival in a pancreatic cancer cells and macrophages co-culture system.

To address the rigor noted for Fig 6d (now Fig 6I), KC tumor content of FPN and F4/80 colocalization was quantified and new graph is added to the right of 6i showing Ferrichrome downregulated FPN on macrophages.

Comment # 9: *What is the relative importance of macrophage-mediated phagocytosis of cancer cells, vs. other mechanisms including NK or CD8 T cell mediated killing? From my knowledge,*

macrophage-mediated phagocytosis is not a primary means of anti-tumor immunity. Does blocking macrophage-mediated phagocytosis lead to significant tumor progression?

Response: This is a very good question since typically cytotoxic T cell killing is considered the primary antitumor function to reduce tumor progression (PMID: 32509781). We have addressed your concern by providing support from the literature demonstrating the importance of macrophage-mediated phagocytosis on tumor progression PMID: 28424516, PMID: 30664738, PMID: 20813259, PMID: 23690610. For instance, blockade of CD47 (which is a “do-not-eat me signal”) in cancer cells synergizes with rituximab to promote phagocytosis and eradicate non-Hodgkin lymphoma (PMID: 20813259). Additionally, anti-CD47 antibody-mediated phagocytosis of cancer cells by macrophages drives a T-cell-mediated antitumor immune response by priming CD8 T cells (PMID: 23690610). Therefore, we have edited our discussion to emphasize that cancer cell phagocytosis by macrophages is an important process in antitumor immunity acting like a double hit by directly eliminating cancer cells, and also by priming CD8 T cells. Our results support this by demonstrating that ferrichrome led to an increase in both CD8 T cell frequencies in KC tumors (Fig 7D) in addition to elevations in phagocytosis (above and figure 3D). The discussion has been edited and these references are now included (**lines 370-374**).

Comment # 9: *The sections on PD-L1, phagocytosis and killing, as well as cancer cell invasion and migration, are mainly correlative. It's unclear how these observations explain the in vivo results that ferrichrome prevents tumor growth in the subcutaneous model. Furthermore, it is difficult to envision how these various pathways and macrophage activities may be related and contribute to anti-tumor effects.*

Response: We regret that we were not clearer about macrophage-mediated antitumor impacts on tumor progression. In the previous answer, we have detailed the role of phagocytosis and direct macrophage killing as well as impacts on priming CD8 T cells in antitumor immunity. In addition, in supplementary figures 6A and 6D, we reported that ferrichrome treatment increased PD-L1 expression in whole tumor and also on macrophages. PD-L1 expression on macrophages positively correlates with infiltration of CD8 T cells. Consequently, high levels of macrophage PD-L1 is associated with greater immunotherapy efficacy and longer survival in cancer patients (PMID: 31615933). Therefore, some of the antitumor effects observed in the ferrichrome + anti-PD-L1 combination group may be attributed to higher PD-L1 expression in the tumor, potentially through macrophage PD-L1 in particular. Finally, tumor associated macrophages promote cancer cell migration and invasion which leads to metastasis and advanced disease severity (PMID: 32509781). We show that ferrichrome treatment of macrophages significantly decreases their capacity to promote cancer cell migration and invasion (Fig 4A-C). While our *in vitro* work supports that ferrichrome impacts macrophage capacity to promote cancer cell migration, invasion and aggressiveness, additional studies to determine the contribution of macrophage-specific effects are needed and planned in future studies. To address reviewer concern, we have incorporated these comments into our discussion and softened our language to indicate that while our results are supportive for macrophage-mediated impacts on ferrichrome-induced reduction of tumor

progression, other cells or the crosstalk between macrophages and T cells for example could be the primary mediators of antitumor function (lines 382-385, 396-405).

Comment # 10: *Figure 7 begins to explore an anti-tumor mechanism involving CD8 T cells, but goes no further than showing T cell numbers and reduced tumor volume when Ferrichrome is combined with anti-PD-L1. I'm unsure what this figure contributes to our mechanistic understanding of anti-tumor immunity by ferrichrome. Importantly, anti-PD-L1 does not significantly reduce tumor volume, and the reduction by ferrichrome plus anti-PD-L1 does not appear different from ferrichrome alone (Fig 1). Thus ferrichrome does not increase the efficacy of anti-PD-L1 treatment, as claimed.*

Response: We thank the reviewer for this astute observation. It is true that anti-PD-L1 treatment is ineffective in reducing tumor progression in this model (now shown as individual mice in Fig 7h for each treatment). We in fact capitalized upon this finding and chose this approach as an immunotherapy-resistant model to determine if ferrichrome will improve therapeutic efficacy. We regret that we failed to include proper controls in this figure in the previous submission. We have revised the current figure to include both vehicle and ferrichrome treated mice in addition to IgG, PD-L1- and combination therapy Ferrichrome+PD-L1. While Ferrichrome alone did reduce tumor progression (as you noted is similar to previous experiments reported in Fig 1D) and shown now as individual mice in 7h, ferrichrome did not ameliorate progression or induce regression. However, as we had previously shown, the combined effect of ferrichrome plus anti-PD-L1 improved PD-L1 efficacy. To address your concern and to avoid overstating findings, we included the proper controls and edited methods, legend, results and discussion. In sum, our results demonstrate that ferrichrome appears to improve PD-L1 efficacy. However, the role of ferrichrome alone, including dose and route of delivery are worthy of future studies.

Comment #11: *Fig 2a is confusing to me – if M2 markers are reduced, why does the end of the pathway indicate M2>M1?*

Response: We regret this typo. We have corrected it to say “M1>M2”.

Comment #12: *Fig 2b: TGF-B is not an M1 marker*

Response: We regret this typo and have fixed it to say M2 marker.

Comment #12: *Fig 2 and others: Why are different M1/M2 markers used for human vs. mouse?*

Response: We regret this mistake. Gene names were corrected Fig 2 B and C. Only mouse primers were used. Most markers were the same in RAW264.7 cells and KC tumor for M2 and M1 determination.

Comments of Reviewer #2:

The current study by Chaib et al investigated the anti-tumorigenic effects of ferrichrome (Lactobacillus casei derived siderophore) on a murine pancreatic cancer model. Specifically,

intratumoral administration of ferrichrome reprogramed tumor-associated macrophages (TAMs), increased CD8+ T cell infiltration into tumors and markedly reduced tumor burden. The authors made an interesting observation that altered immune response by ferrichrome improved anti-PD-L1 therapy. The authors also show that ferrichrome can induce TAMs polarization via activation of the TLR4 pathway that represses the expression of iron export protein ferroportin (FPN1) in macrophages. The results are intriguing, however there are some major concerns which should be addressed.

Comment #1: *Although the data suggests that ferrichrome is able to activate TLR4, no direct evidence has been provided. Authors need to demonstrate ferrichrome's ability to induce classical pro-inflammatory cytokines of TLR4 pathway in vivo using WT and TLR4-deficient mice. Alternatively, authors could use the commercially available reporter cell lines, like TLR4-expressing HEK293 cells, to substantiate ferrichrome-TLR4 interaction.*

Response: We appreciate and thank for reviewer comment. While we believe that by using TLR4 deficient macrophages, we have demonstrated that TLR4 is necessary for Ferrichrome's actions in both M1 and M2 cytokine expression (existing data shown in Fig 6F). We reported that ferrichrome reduced M2 markers and elevated M1 markers, which was absent in TLR4^{-/-} macrophages. Additionally, we had utilized the TLR4 inhibitor CLI-095 (InvivoGen) in Figure 6g and 6h (PMID: 32367494, PMID: 31852918) which is well-established pharmacological inhibitor to inhibit TLR4 action through a specific inhibition of the MD2 region on TLR4 (PMID: 18299127). Based on similarities in the chemical structure, we estimated that ferrichrome would signal in that same moiety. Yet, you are correct in that we have not shown a direct Ferrichrome-TLR4 interaction such that we may claim ferrichrome is a novel ligand for TLR4 and have ensured that we make no such claim in the manuscript of a direct effect. To additionally address your concern, we took your suggestion into account to use WT and TLR4 KO mice and showed that ferrichrome acts through TLR4 to shift macrophage polarization *in vivo* similar to *in vitro* findings. Please refer to **new in vivo** figures **Fig. 6A-E** and **Supplementary Fig. 10** wherein we demonstrate that ferrichrome treatment increased MHCII^{hi} macrophages and reduced MHCII^{lo} macrophages in the spleen, but this effect was lost in TLR4^{-/-} mice.

Comment #2: *The source of ferrichrome is not provided. What is the purity of ferrichrome? Any endotoxin contamination? Is it iron free? Most importantly, authors need to perform some of the critical in vitro experiments in the presence of Polymyxin B. These need to be rigorously addressed.*

Response: We regret not providing these critical details. We have included the company and additional details on the Ferrichrome we purchased in Methods on lines **452-456**. Ferrichrome that we purchased is iron-free and was isolated and purified from *Ustilago sphaerogena* (**Sigma F8014**). In addition, we tested ferrichrome for endotoxin as detailed above (**Fig. S3**). We believe that the reviewer is suggesting to use Polymyxin B as an antimicrobial. Because we aim to use ferrichrome as a drug and *in vivo* and *in vitro* data support a direct role of ferrichrome, we believe that studies examining the impact of microbiota and antibiotic treatments such as polymyxin B is beyond the scope of this

study. While it is established that the microbiome impacts cancer progression as well as chemo- and immuno-therapies, it is beyond the scope of this manuscript.

Comment #3: *It is not clear whether the iron chelation property of ferrichrome exert effects on the tumor model. Authors stated that: “Moreover, we elucidated that ferrichrome regulates macrophage iron metabolism by decreasing Fpn expression in a TLR4-dependent manner, not by iron chelation” (lines # 113-115). However, the authors measured ferritin as an indirect marker, but did not use methodology that directly assess changes in the intracellular iron levels in ferrichrome treated cells.*

Response: We thank reviewer 2 for this insightful comment. We agree that while we reported an increase in ferritin (iron sequestration and storage), this is an indirect marker. In addition, we had demonstrated a ferrichrome-mediated reduction in iron in the TME (supplemental fig 7A) by Prussian blue staining. However, you are correct that we have not yet shown that ferrichrome alters intracellular iron levels in macrophages. To address your concerns, intracellular iron levels were assessed in M0, M1 (LPS-treated) and ferrichrome-treated BMDMs using an iron detection kit (Abcam). Results demonstrated supportive evidence that in the presence of ferrichrome, iron levels increased in BMDM (suggesting elevated iron sequestration) which is also an indicative of the M1 phenotype (**Please refer to new Supplementary Fig. 9 a images and b quantification**).

Comment #4: *Is this anti-tumoral property is unique to ferrichrome? Or any other water soluble siderophores (e.g. DFO) can do? Addressing this point can provide insight on whether the anti-tumor effects are independent of iron chelation.*

Response: This is an important question. To determine if the effect of iron chelation by siderophores is general or specific to ferrichrome, we turned to the literature to address this concern. Other studies have shown that iron treatment of macrophages with iron chelators (such as deferoxamine) decrease macrophage production of proinflammatory cytokines such as IL-1 β , TNF α and do not favor the M1 phenotype (PMID: 31291584). We believe that this anti-tumoral property is specific to ferrichrome-TLR4 signaling, not necessarily iron chelation because of supportive data in figure S10 and new data in figure 6. Therefore, while we have not tested other siderophores, we posit that those that may interact with TLR4 such as ferrichrome, may have similar effects to be tested in future studies. In sum, using TLR4 KO macrophages in Fig 6F combined with TLR4 pharmacologic inhibitor that is well established to blunt TLR4 signaling (Fig 6G), we showed that dual loss of function approaches (genetic and pharmacological) completely abrogated the proinflammatory macrophage phenotype (M1-like) induced by ferrichrome. We have also confirmed this effect *in vivo* using TLR4 KO mice (new experiments shown in Fig. 6A-E, discussed above). In other words, we show that the M1 phenotype induced by ferrichrome is TLR4-dependent. Responses to questions to reviewer 1 additionally show TLR4 dependency by ferrichrome in inducing M1 phenotype in macrophages.

Comment #5: *The authors did not show anti-tumorigenic effects of Lactobacillus casei. Whether L. casei (heat killed or live) in combination would improve the efficacy anti-PD-L1?*

Response: We regret this mistake and thank reviewer 2 for pointing it out. We have removed this comment about the potential anti-tumor role of the microbe *L. Casei* because the intention of ferrichrome as presented herein is as a pharmaceutical delivered with immunotherapy not as a microbial modification per se. Examining the impacts of the microbiome is important, as mentioned in answer above, but beyond the scope of this manuscript. To address your concern, we edited the discussion to indicate that while microbes are important in regulating the immune milieu and therapy in patients (PMID: 32320071) further studies are needed to determine the impact of microbes on pancreatic cancer.

Comment #6: *Supplementary Figure: 2A, lower panel marked wrongly, it should be M1>M2.*

Response: We regret this typo. We have corrected it to say “M1>M2”

We thank the reviewers for their constructive comments and because of these critiques, our manuscript has significantly improved.

Reviewers' comments:

Reviewer #1 (Remarks to the Author):

Reviewer #1 response to revisions:

The authors present a very interesting and potentially important story, however, there are still questions regarding rigor and data interpretation. The revisions do not fully address my previous concerns. I comment on each below.

2. Role of LPS: The authors have measured endotoxin concentrations in their Ferrichrome batch (0.017 EU/ml), which shows that the levels are above water, but around the lowest dose of standard (0.01 EU/ml).

3. Role of iron in ferrichrome's effects: The authors measure intracellular iron levels in macrophages. This is insufficient. First, total iron is not the same as available iron – ferrichrome is a chelator therefore available iron is important. Moreover, to demonstrate that iron is not involved, one should test if addition/depletion of iron impacts the phenotypes, not simply measure iron. Most importantly, Suppl figure 7 shows there is indeed an effect of iron on the ferrichrome-induced effects! This revision does not address my original concern.

4. Impact of ferrichrome on other cancer cells, as in Konishi et al: The authors have tested two additional cancer cell lines – one of which was tested in Konishi et al. They conclude that there is a moderate effect on viability for the cancer cell lines, but no effect on RAW cells. This interpretation is flawed – the viability of cancer cell lines is 60-80% after 10ug/ml treatment, but only 45-60% after the same treatment in RAW cells. Furthermore, there is no explanation to why their results differ from Konishi et al. Therefore this concern is not sufficiently addressed.

5. Induction of apoptosis: The authors measure apoptosis and conclude it is not induced in their cell lines, nor in SW620 that was tested in Konishi et al. Why are these results in SW620 the opposite of what was found in Konishi et al? This revision does not address my concern.

6. Normalization M2 markers to IL-4-untreated control: The authors show that their IL-4 treatment induces M2 markers, but it also induces M1 markers. Therefore, it is not fully convincing that these cells are truly M2 skewed. More importantly, this untreated baseline is provided in supplemental and not in the main figure... When one qualitatively compares the results between these figures, an alternative interpretation arises that perhaps the macrophages are not more M1 skewed, but perhaps just did not skew towards M2. These new data do not demonstrate that cells are M2. Also note that there is a typo in figure S5a – the bottom row should indicate M1>M2.

7. Total macrophages: This was already addressed

8. Lack of rigor:

The authors have now quantified images shown in 2D, which shows a non-significant reduction in co-localization. Fig 2E is added but shows only 1 representative flow plot. Suppl fig 5 attempts to better characterize these macrophages, also in the presence of UN-KC-6141 cells. I cannot provide a positive comment on the rigor of these experiments, as it is unclear how many technical and biological replicates were performed, but likely indicating a very small n# due to the unusually error in SF5c for qPCR analysis. This concern is not addressed.

Regarding Figure 3D, there is still no quantification of in vivo phagocytic events, yet the authors conclude there are more in the ferrichrome treated tissue. Data do not yet support this. These co-localization data could be quantified, as in 6i. Furthermore, there are alternative interpretations to

figure 3c, as this is not a phagocytosis assay but a killing assay. The two are independent actions of a macrophage.

Co-localization in 6i has been quantified, but shows a non-significant difference that is interpreted as being a reduction.

Comments 9 and 10 address correlations, mechanism, etc.:

The authors have attempted to address these concerns and clarify intention and data interpretation. They still do not fully highlight what specific mechanism is driving the overall observations that ferrichrome treatment reduces pancreatic cancer.

Reviewer #2 (Remarks to the Author):

Authors have addressed all my concerns.

Now, authors have clearly mentioned the source of ferrichrome i.e., *Ustilago sphaerogena*, which is a fungus. Therefore, it is meaningless to include SFig.3. and should be removed.

Dear Dr. Moroishi,

We thank the reviewers for their constructive comments and because of these critiques, our manuscript has significantly improved. We have addressed all the comments of reviewer #1. Below is our point-by-point response and changes in the revised manuscript are highlighted in yellow text.

Comments of Reviewer #1:

Comment # 1: *The authors present a very interesting and potentially important story, however, there are still questions regarding rigor and data interpretation. The revisions do not fully address my previous concerns. I comment on each below.*

Response: We thank reviewer 1 for the comments and suggestions which have potentially improved our highly impact manuscript.

Comment # 2: *Role of LPS: The authors have measured endotoxin concentrations in their Ferrichrome batch (0.017 EU/ml), which shows that the levels are above water, but around the lowest dose of standard (0.01 EU/ml).*

Response: It was a typographical error. The endotoxin level in Ferrichrome batch was measured 0.0074 EU/ml which is lower than the endotoxin concentration in water and lowest dose of standard (0.01 EU/ml). We have corrected it in our revised supplementary figure 3.

Comment # 3: *Role of iron in ferrichrome's effects: The authors measure intracellular iron levels in macrophages. This is insufficient. First, total iron is not the same as available iron – ferrichrome is a chelator therefore available iron is important. Moreover, to demonstrate that iron is not involved, one should test if addition/depletion of iron impacts the phenotypes, not simply measure iron.*

Response: We thank reviewer 1 for their insightful comments. When we measured intracellular iron in supplementary figure 8, we first treated cells with either LPS or ferrichrome for 48h, then we made sure all ferrichrome or LPS was removed by washing with PBS twice before the addition of the same concentration of iron to all samples. Therefore, available iron was the same to all samples while we made sure that ferrichrome was not present in the supernatant when we added iron because ferrichrome is an iron chelator. As per the reviewer 1 suggestion, we have performed additional experiments to determine the effect iron alone, iron + ferrichrome, iron + LPS, iron + ferrichrome + LPS on the phenotype of BMDMs using flow cytometry analysis. In this experiment, we treated BMDMs with above mentioned treatments and BMDMs phenotypes were measured by analyzing the M1 markers (CD86 and MHCII); and same treatment groups except that LPS was replaced with IL-4 to measure the M2 marker CD206. Previous studies have shown that iron skews macrophage polarization toward the M1 phenotype (PMID: 30835899, PMID: 29167669). Our findings were consistent with these studies as we found that iron treatment increased MHCII expression and

decreased CD206 expression in BMDMs (see new supp Fig. 8 C-E). However, iron did not increase the M1 marker CD86. Importantly, iron + ferrichrome + LPS had increased M1 markers compared to iron + LPS while iron + IL-4 + ferrichrome had decreased M2 marker CD206 compared to iron + IL-4. These results confirm that ferrichrome decreases M2 markers and increases M1 markers in the presence of iron and stimuli which better mimics *in vivo* conditions (iron here was also removed from cell culture supernatants before the addition of other treatments to prevent chelation effect of ferrichrome). We have added these findings to supplementary figure 8 and results section.

Comment # 4: *Most importantly, Suppl figure 7 shows there is indeed an effect of iron on the ferrichrome-induced effects! This revision does not address my original concern.*

Response: We regret that we were unable to clearly articulate regarding our findings in supp fig. 7A (Now supplementary Figure 6A). Images shown in Supp fig. 6A are from whole tumor sections where iron was quantified. This is total iron available in the tumor microenvironment and not only iron available to macrophages. It is important to note that availability of iron to cancer cells helps cancer cells proliferation which is an opposite effect we see in macrophages. When macrophages are skewed toward the M1 phenotype, they typically up-regulate ferritin to store iron and downregulate the iron export protein ferroportin. Cancer cells behave similarly by trying to sequester iron to their advantage. It is possible that the decrease in iron levels seen in whole tumor in supp fig. 6A is not solely due to ferrichrome's chelation effect, but to the modified intratumoral immune landscape where macrophages may sequester iron and not export it to cancer cells because they are now more skewed toward a proinflammatory phenotype. We certainly appreciate that this is the topic of great interest and warrants deep investigations in future to understand this point better. We again appreciate your valuable feedback.

Comment # 4: *Impact of ferrichrome on other cancer cells, as in Konishi et al: The authors have tested two additional cancer cell lines – one of which was tested in Konishi et al. They conclude that there is a moderate effect on viability for the cancer cell lines, but no effect on RAW cells. This interpretation is flawed – the viability of cancer cell lines is 60-80% after 10ug/ml treatment, but only 45-60% after the same treatment in RAW cells. Furthermore, there is no explanation to why their results differ from Konishi et al. Therefore this concern is not sufficiently addressed.*

Response: We regret this error in interpretation. Reviewer 1 is correct that these doses cause a decrease in proliferation at 48 hours comparable between RAW and cancer cells. We have edited the manuscript to reflect this. We added a comment about reasons why Konishi and our work may be different. The slight variance which is apparent in our data might be due to difference in cell passage number and/or clonal variation. Fig.S1 is MTT assays. However, in Fig S2 we did apoptosis analysis by annexin staining and showed no effect. Therefore the 20% decrease measured by MTT may be an impact on proliferation but is likely not caused by apoptosis.

Comment # 5: *Induction of apoptosis: The authors measure apoptosis and conclude it is not induced in their cell lines, nor in SW620 that was tested in Konishi et al. Why are these results in SW620 the opposite of what was found in Konishi et al? This revision does not address my concern.*

Response: It may be variances in cell passage number, clonal variations, or other unknown cell growth dynamics. We provided a detailed methodology on how our experiments were conducted and can only confidently comment on our findings.

Comment # 6. *Normalization M2 markers to IL-4-untreated control: The authors show that their IL-4 treatment induces M2 markers, but it also induces M1 markers. Therefore, it is not fully convincing that these cells are truly M2 skewed.*

Response: The term M1 and M2 were based on initial *in vitro* description of a few genes but, the M1 and M2 phenotypes are often overlapping (Orecchioni et al., PMID: 31178859), therefore M1 and M2 markers often overlap but the overwhelming majority of M2 markers are upregulated, therefore we are confident with calling them M2. Additionally RAW cells are inherently M1 skewed. Moreover, to account for known hindrances/hurdles/ problems with RAW cells, we used peritoneal macrophages and confirmed findings *in vivo* as well.

Comment # 6. *More importantly, this untreated baseline is provided in supplemental and not in the main figure. When one qualitatively compares the results between these figures, an alternative interpretation arises that perhaps the macrophages are not more M1 skewed, but perhaps just did not skew towards M2.*

Response: In response to the previous comment, we had provided this information in supplementary Figure 3. Now, as per reviewer's suggestion, we have included these results in main figure 2. IL-4 and IL-4 + ferrichrome groups are normalized to the control.

Comment # 7. *These new data do not demonstrate that cells are M2.*

Response: We respectfully disagree with the reviewer comment, as we observed significantly increased expression of five to six M2 markers upon IL-4 treatment. As described in previous response of comment 6, the new graph comprising all treatment groups has been incorporated in figure 2B.

Comment # 8. *Also note that there is a typo in figure S5a – the bottom row should indicate M1>M2.*

Response: We have fixed this typographical error in the revised figure (Now S Fig. 4a).

Comment # 9. *Total macrophages:*

Response: Total macrophages counts is provided in **Supplementary Fig. 5**.

Comment # 10. *Lack of rigor: The authors have now quantified images shown in 2D, which shows a non-significant reduction in co-localization.*

Response: We appreciate reviewer's suggestion, as it is very difficult to acquire co-localization data in absolute values, we have revised our text to indicate that there was a

remarkable reduction and deleted the word “significant” from the text. However, we showed this in other complementary ways including 2e and TLR4 new data Fig. 6E and sup fig. 10. Therefore, this does not change our conclusion.

Comment # 11. *Fig 2E is added but shows only 1 representative flow plot.*

Response: The representative flow plot shows another M2 marker which is CD206. Just like we mentioned in the previous point, our findings were consistent showing that ferrichrome decreases M2 markers both *in vitro* and *in vivo* (see Fig 1G-H with CD163, Fig. 2, Fig 6E,F, Fig S5B, Fig S10). Therefore, we proved this point multiple times in different experimental settings and are confident about our findings.

Comment # 11. *Suppl fig 5 attempts to better characterize these macrophages, also in the presence of UN-KC-6141 cells. I cannot provide a positive comment on the rigor of these experiments, as it is unclear how many technical and biological replicates were performed, but likely indicating a very small n# due to the unusually error in SF5c for qPCR analysis. This concern is not addressed.*

Response: *We assume reviewer is referring to supplementary figure 5E as this is qPCR, co-culture experiment. Since we have shown multiple times that ferrichrome alters macrophage polarization in vitro and in vivo, we believe this experiment doesn't add to the story so we removed it to alleviate reviewers concerns.*

Comment # 12. *Regarding Figure 3D, there is still no quantification of in vivo phagocytic events, yet the authors conclude there are more in the ferrichrome treated tissue. Data do not yet support this. These co-localization data could be quantified, as in 6i.*

Response: We used techniques for *in vivo* phagocytic events based on previous literature (PMID: 30664738) and we provided complementary phagocytic assays to support this. To address the concerns about lack of quantification, we could not co-localize as we typically do for IF studies because these do not overlap- one color is within the other color. Therefore, we manually counted phagocytic events which we defined as red (CK19) inside green (F4/80) and quantified these using ImageJ. New data is presented in the main figure 3C.

Comment # 13. *Furthermore, there are alternative interpretations to figure 3c, as this is not a phagocytosis assay but a killing assay. The two are independent actions of a macrophage.*

Response: We agree with the reviewer's comment and have modified the language in the revised results section of the manuscript.

Comment # 13. *Co-localization in 6i has been quantified but shows a non-significant difference that is interpreted as being a reduction.*

Response: This is accurate that the quantification is not significant therefore we altered our language. However, in macrophages we show *in vitro* that Ferrichrome decreases

FPN and supplemental Fig. 8. One reason may be that for IF, we used a broad macrophage marker F4/80 and this doesn't capture the M1/M2 skewing as well as other markers. In future studies, we will do more advanced IF analyses for subtype co-localization. Comments 9 and 10 address correlations, mechanism, etc.

Comment # 14. *The authors have attempted to address these concerns and clarify intention and data interpretation. They still do not fully highlight what specific mechanism is driving the overall observations that ferrichrome treatment reduces pancreatic cancer.*

Response: To summarize our study, we found that ferrichrome skews macrophage polarization *via* TLR4 and promotes an antitumor phenotype of macrophages. Although we showed that ferrichrome increases macrophage phagocytic capacity both *in vitro* and *in vivo*, thus it be one of the major mechanisms for its anticancer activity in pancreatic cancer as this the main immune cell population in tumors. However, it will be interesting to investigate the impact of ferrichrome on other immune cell populations to fully elucidate mechanisms that are responsible for ferrichrome's antitumor effect. It is likely that ferrichrome remodels the TME *via* impairing macrophage polarization toward an M2 phenotype as evidenced by increased CD8 T cell infiltration in ferrichrome-treated tumors. It is also possible that ferrichrome acts *via* other cells of the innate immune compartment such as MDSCs and DCs which will be the subject of future studies. We have altered our discussion to emphasize on the above-mentioned points and we agree that future studies may reveal alternative mechanisms of ferrichrome's antitumor effect.

Comments of Reviewer #2:

Comment: *Authors have addressed all my concerns. Now, authors have clearly mentioned the source of ferrichrome i.e., *Ustilago sphaerogena*, which is a fungus. Therefore, it is meaningless to include SFig.3. and should be removed.*

Response: We thank second reviewer's comments and suggestions.

Overall, we are highly obliged for the meticulous and thorough reviews of our manuscript and comments/suggestions provided by both extremely knowledgeable reviewers (1 and 2). Their comments/suggestions have certainly improved the quality of our manuscript. We strongly believe that this is a much stronger manuscript now, and its timely publication will advance pancreatic cancer treatment field in the future.

Dear Dr. Moroishi,

We thank all the reviewers and the editorial board members for providing constructive criticisms and taking our response to consider our manuscript. We have edited our manuscript according to the journal guidelines and changes in the revised manuscript are highlighted in yellow text. Following specific changes were made in the manuscript.

1. All the figures were made in 300 dpi resolution and uploaded as TIFF files.
2. Supplementary data of each main and supplementary figures was provided in XL files.
3. Data availability section was provided in the revised article file.